# X-ray quasi-periodic eruptions from two previously quiescent galaxies

R. Arcodia[1✉], A. Merloni[1], K. Nandra[1], J. Buchner[1], M. Salvato[1], D. Pasham[2], R. Remillard[2], J. Comparat[1], G. Lamer[3], G. Ponti[1,4], A. Malyali[1], J. Wolf[1], Z. Arzoumanian[5], D. Bogensberger[1], D. A. H. Buckley[6], K. Gendreau[5], M. Gromadzki[7], E. Kara[2], M. Krumpe[3], C. Markwardt[5], M. E. Ramos-Ceja[1], A. Rau[1], M. Schramm[8] & A. Schwope[3]

Quasi-periodic eruptions (QPEs) are very-high-amplitude bursts of X-ray radiation recurring every few hours and originating near the central supermassive black holes of galactic nuclei[1,2]. It is currently unknown what triggers these events, how long they last and how they are connected to the physical properties of the inner accretion flows. Previously, only two such sources were known, found either serendipitously or in archival data[1,2], with emission lines in their optical spectra classifying their nuclei as hosting an actively accreting supermassive black hole[3,4]. Here we report observations of QPEs in two further galaxies, obtained with a blind and systematic search of half of the X-ray sky. The optical spectra of these galaxies show no signature of black hole activity, indicating that a pre-existing accretion flow that is typical of active galactic nuclei is not required to trigger these events. Indeed, the periods, amplitudes and profiles of the QPEs reported here are inconsistent with current models that invoke radiation-pressure-driven instabilities in the accretion disk[5–9]. Instead, QPEs might be driven by an orbiting compact object. Furthermore, their observed properties require the mass of the secondary object to be much smaller than that of the main body[10], and future X-ray observations may constrain possible changes in their period owing to orbital evolution. This model could make QPEs a viable candidate for the electromagnetic counterparts of so-called extreme-mass-ratio inspirals[11–13], with considerable implications for multi-messenger astrophysics and cosmology[14,15].

Given its large collecting area and blind survey strategy, the eROSITA instrument on the Spectrum-Roentgen-Gamma (SRG) space observatory[16] is capable of systematic searches for X-ray sources that are variable on timescales of hours to months (see Methods for more details). This applies to QPEs, which thus far have only been detected in X-rays[1,2]. The first QPE observed by eROSITA, hereafter eRO-QPE1, showed high-amplitude X-ray variability within just a few hours. It showed a strong X-ray signal in two eROSITA survey scans that were preceded, separated and followed by scans in which the signal was much fainter (Fig. 1a). Similar to the two previously known QPE sources—GSN 069[1] and RX J1301.9+2747[2]—the X-ray spectrum is very soft with most of the counts originating from below approximately 1.5–2 keV and consistent with a thermal black-body emission. As with the light curve, the spectrum shows oscillations from a faint to a bright phase (Fig. 1b). We identify eRO-QPE1 as originating within the nucleus of the galaxy 2MASS 02314715-1020112, for which we measured a spectroscopic redshift of $z = 0.0505$ (see Methods section 'The host galaxies of the QPEs'). The related eROSITA quiescence ($1\sigma$ upper limit) and peak intrinsic 0.5–2-keV luminosities are $<2.1 \times 10^{41}$ erg s$^{-1}$ and approximately $9.4 \times 10^{42}$ erg s$^{-1}$, respectively, if the X-ray spectra are modelled with a standard accretion disk model (see Methods section 'X-ray spectral analysis').

Two follow-up observations triggered with the XMM-Newton X-ray telescope confirmed the remarkable bursting nature of the source (Fig. 1c, d). The first observation (hereafter eRO-QPE1-XMM1) found the source in a faint state for approximately 30 ks, followed by a sequence of three consecutive asymmetric bursts, possibly partially overlapping (Fig. 1c), which is behaviour that has not been previously observed in QPEs[1,2]. In terms of intrinsic 0.5–2-keV luminosity, after an initial quiescent phase at about $2.3 \times 10^{40}$ erg s$^{-1}$ the first burst was characterized by a fast rise and slower decay lasting around 30 ks and peaking at approximately $3.3 \times 10^{42}$ erg s$^{-1}$; it was then followed by a second fainter burst (peak at approximately $7.9 \times 10^{41}$ erg s$^{-1}$) and by a third, which was the brightest (peak at approximately $2.0 \times 10^{43}$ erg s$^{-1}$) but was only caught during its rise. The second XMM-Newton observation (hereafter eRO-QPE1-XMM2) showed an eruption very similar to the first seen in eRO-QPE1-XMM1 in terms of amplitude and luminosity, although lasting for >40 ks, that is, for almost as much as the three in eRO-QPE1-XMM1 combined (Fig. 1c). To better characterize the physics and to determine the duty cycle of these eruptions, we started an intense monitoring campaign with the NICER X-ray instrument aboard the International Space Station (ISS), which revealed 15 eruptions in about 11 days (Fig. 1d).

[1]Max-Planck-Institut für Extraterrestrische Physik, Garching, Germany. [2]MIT Kavli Institute for Astrophysics and Space Research, Cambridge, MA, USA. [3]Leibniz-Institut für Astrophysik Potsdam (AIP), Potsdam, Germany. [4]INAF—Osservatorio Astronomico di Brera, Merate, Italy. [5]Astrophysics Science Division, NASA Goddard Space Flight Center, Greenbelt, MD, USA. [6]South African Astronomical Observatory, Cape Town, South Africa. [7]Astronomical Observatory, University of Warsaw, Warsaw, Poland. [8]Graduate School of Science and Engineering, Saitama University, Saitama, Japan. ✉e-mail: arcodia@mpe.mpg.de

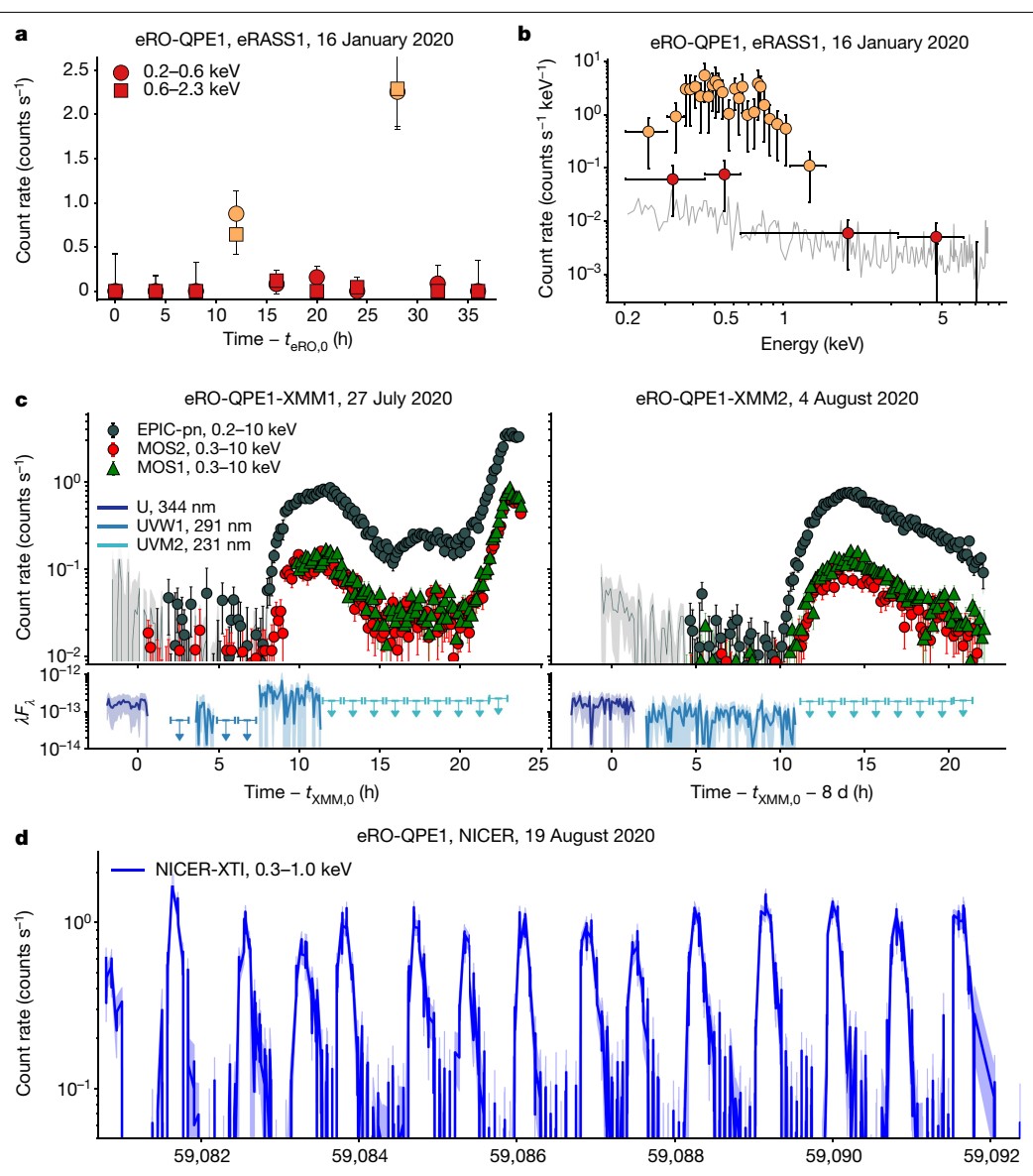

**Fig. 1 | The first eROSITA QPE.**
**a**, eROSITA light curve in the 0.2–0.6-keV and 0.6–2.3-keV energy bands (circles and squares, respectively), with red and orange highlighting faint and bright observations, respectively. The start of the light curve $t_{eRO,0}$ is approximately MJD 58864.843 (MJD, modified Julian date), observed during the first eROSITA all-sky survey (eRASS1). **b**, eROSITA X-ray spectra of the bright and faint states in orange and red, as in **a**. **c**, Background-subtracted XMM-Newton X-ray light curves with 500-s bins for the European Photon Imaging Camera (EPIC) instruments: EPIC-pn (dark grey), MOS1 (green) and MOS2 (red) in the energy band shown in the legend. The beginning of both observations was contaminated by flares in the background and excluded; the dark grey solid line and contours show the underlying ≤1-keV EPIC-pn light curve to give a zeroth-order extrapolation of the rate, excluding the presence of obvious soft X-ray eruptions. $t_{XMM,0}$ corresponds to the start of the cleaned MOS2 exposure in the first observation, approximately MJD 59057.805. XMM-Newton optical and UV fluxes are shown in the lower panels (units of erg cm$^{-2}$ s$^{-1}$, where $F_\lambda$ is the spectral flux density and $\lambda$ is the wavelength in angstroms), with non-detections shown as upper limits. **d**, Background-subtracted NICER-XTI light curve. The mean (and dispersion-on) rise-to-decay duration is approximately 7.6 h (~1.0 h) and the peak-to-peak separation is approximately 18.5 h (~2.7 h). All uncertainties are 1σ, shown as error bars or shaded regions.

The second eROSITA QPE we have detected, hereafter eRO-QPE2, showed similar variability patterns and X-ray spectra as eRO-QPE1 during the X-ray sky survey (Fig. 2a, b). We associated it with the galaxy 2MASX J02344872-4419325 and determined a spectroscopic redshift of $z = 0.0175$ (see Methods section 'The host galaxies of the QPEs'). The related intrinsic 0.5–2-keV luminosities of the quiescent (1σ upper limit) and peak phases are $<4.0 \times 10^{40}$ erg s$^{-1}$ and approximately $1.0 \times 10^{42}$ erg s$^{-1}$, respectively. A follow-up observation with XMM-Newton revealed nine eruptions in a single day, oscillating between approximately $1.2 \times 10^{41}$ erg s$^{-1}$ and $1.2 \times 10^{42}$ erg s$^{-1}$ in the 0.5–2-keV band (Fig. 2c). In neither eRO-QPE1 nor eRO-QPE2 is there evidence of simultaneous optical/UV variability (see Figs. 1c, 2c), in agreement with the behaviour of GSN 069[1].

eRO-QPE1 shows a distribution of QPE rise-to-decay durations with a mean (dispersion) of approximately 7.6 h (~1.0 h) and a distribution of peak-to-peak separations of about 18.5 h (~2.7 h), as derived from the NICER light curve (Fig. 1d). The duty cycle (mean duration over mean separation) is approximately 41%. Conversely, eRO-QPE2 shows much narrower and more frequent eruptions (see Fig. 2c): the mean (dispersion) of the rise-to-decay duration is approximately 27 min (~3 min), with a peak-to-peak separation of approximately 2.4 h (~5 min) and a duty cycle of around 19%. Interestingly, compared to the two previously known QPEs[1,2], eRO-QPE1 and eRO-QPE2 extend the parameter space of QPE widths and recurrence times towards longer and shorter timescales, respectively. We also note that eRO-QPE1 is the most luminous and the most distant QPE discovered so far, and the most extreme in terms of timescales. The duration and recurrence times of the bursts in eRO-QPE1 are approximately an order of magnitude longer than in eRO-QPE2. This could simply be an effect of the timescales scaling with black hole mass[17]. We estimated the total stellar mass of the two host galaxies and that of eRO-QPE1 is 4–8 times higher than that of eRO-QPE2. Assuming a standard scaling of the black hole mass with stellar mass (see Methods section 'The host galaxies of the QPEs'), this is broadly in agreement with their different X-ray timing properties. Furthermore, peak soft X-ray luminosities of approximately $2 \times 10^{43}$ erg s$^{-1}$ and $10^{42}$ erg s$^{-1}$ for eRO-QPE1 and eRO-QPE2, respectively, exclude a stellar-mass black hole origin, and their X-ray positions, within uncertainties, suggest a nuclear origin (Extended Data Figs. 1a, 2a).

The optical counterparts of eRO-QPE1 and eRO-QPE2 are local low-mass galaxies with no canonical active galactic nuclei (AGN)-like broad emission lines in the optical nor any infrared photometric excess indicating the presence of hot dust (the so-called torus)[18]. In this sense they are similar to GSN 069[3] and RX J1301.9+2747[4], although their optical spectra show narrow emission lines with clear AGN-driven ionization[3,4].

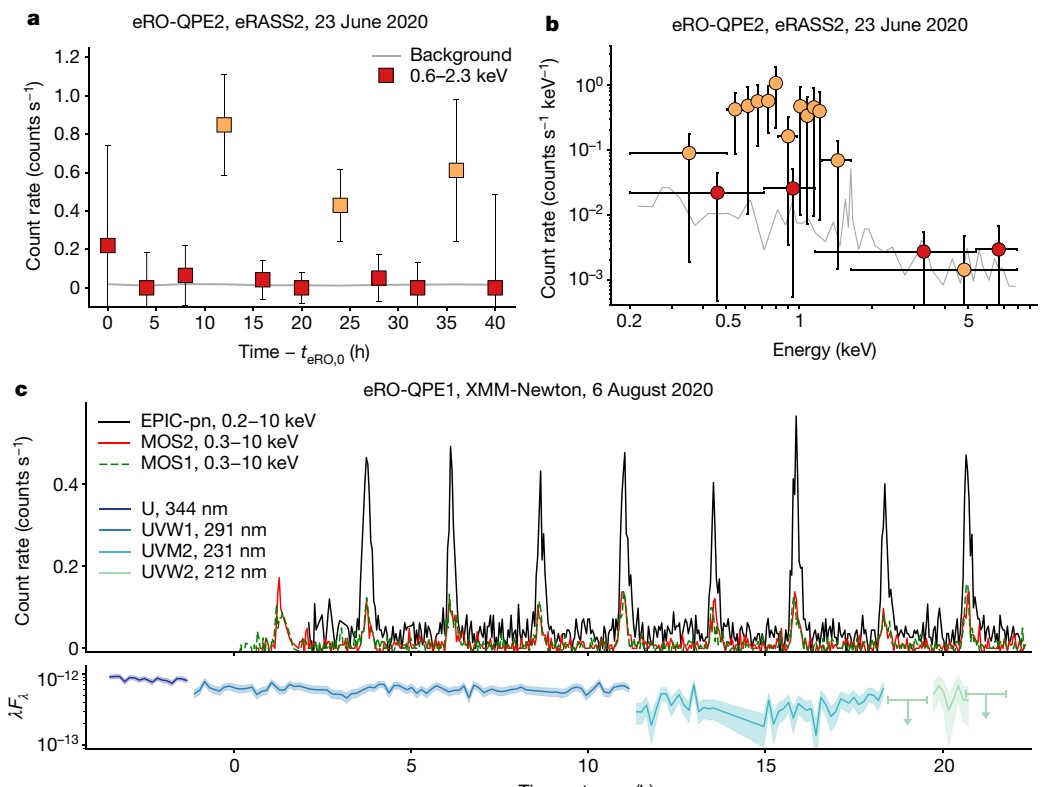

**a** eRO-QPE2, eRASS2, 23 June 2020

**b** eRO-QPE2, eRASS2, 23 June 2020

**c** eRO-QPE1, XMM-Newton, 6 August 2020

**Fig. 2 | The second eROSITA QPE.** **a**, **b**, As in Fig. 1a, b, for eRO-QPE2. The start of the eROSITA light curve is approximately MJD 59023.191. **c**, As in Fig. 1c, for the XMM-Newton observation of eRO-QPE2. $t_{XMM,1}$ corresponds to the start of the cleaned MOS1 exposure, approximately MJD 59067.846. The mean (and related dispersion) of the rise-to-decay duration is about 27 min (~3 min), with a peak-to-peak separation of approximately 2.4 h (~5 min). All uncertainties are $1\sigma$, shown as error bars or shaded regions.

Instead, the optical counterpart of eRO-QPE1 is easily classified as a passive galaxy from the absence of emission lines (Extended Data Fig. 1b), and in eRO-QPE2 the strong narrow emission lines that are observed classify it as a star-forming galaxy (Extended Data Fig. 2b and Methods section 'The host galaxies of the QPEs'). This in turn suggests that the two galaxies reported here have not been active for approximately at least the last $10^3$–$10^4$ years, assuming narrow-line region light-travel timescales[19]. Although the number of known QPEs is too low to reach firm statistical conclusions, our blind search is inherently designed to sample the population of the QPEs' host galaxies without bias, as opposed to serendipitous or archival discoveries, which rely on the source being previously active and known[1,2]. These results hint that perhaps the parent population of QPE hosts consists of more passive galaxies than active galaxies. The X-ray spectra of the QPEs in quiescence are consistent with an almost featureless accretion disk model[1,2] (see Methods section 'X-ray spectral analysis'), although the inactive nature of the host galaxies of our sources argues against a pre-existing AGN-like accretion system.

A few scenarios to explain the QPEs have been suggested[1,10], some based on the presumed active nature of the host black holes of the QPEs. These include so-called limit-cycle radiation-pressure accretion instabilities (see Methods section 'On accretion flow instabilities'), proposed for GSN 069[1] based on the similarities between its observed properties and two extremely variable stellar-mass black holes, namely GRS 1915+105[20,21] and IGR J17091-3624[22]. However, the observed properties of the two QPEs reported here, as well as those of RX J1301.9+2747[2], are inconsistent with the theoretical predictions of this scenario[5-9]. In particular, the faster rise and slower decay of eRO-QPE1 would imply a thicker flow in the cold and stable phase than in the hot and unstable phase, contrary to theory[6]. Moreover, the theory predicts that once the period, the duty cycle and the luminosity amplitude are known, only specific values of black hole mass $M_{BH}$ and viscosity parameter $\alpha$ are allowed[8]: for eRO-QPE1 (eRO-QPE2) one solution is found for $M_{BH} \approx 4 \times 10^6 M_\odot$ ($M_\odot$, solar mass) and $\alpha \approx 5$ ($M_{BH} \approx 3 \times 10^6 M_\odot$ and $\alpha \approx 3$), therefore for the expected masses[1,2] an unphysically high viscosity parameter would be required. Alternatively, more reasonable values of $\alpha \approx 0.1$ and 0.01 would yield very small $M_{BH} \approx 2.4 \times 10^3 M_\odot$ and

$M_{BH} \approx 60 M_\odot$ ($M_{BH} \approx 4.3 \times 10^3 M_\odot$ and $M_{BH} \approx 30 M_\odot$) for eRO-QPE1 (eRO-QPE2). Even in this latter scenario and pushing $\alpha$ as high as approximately 0.2, the resulting thermal timescales for eRO-QPE1 (eRO-QPE2) are $\tau_{th} \approx 20$ s (35 s) at $20 r_g$ ($r_g = GM_{BH}/c^2$, where $G$ is the gravitational constant and $c$ the speed of light in vacuum), which is orders of magnitude smaller than the observed QPE timescales (more details in Methods section 'On accretion flow instabilities').

Extreme or sinusoidal quasi-periodic variability as seen in QPEs is also typically associated with compact object binaries, a scenario which would not require the galactic nuclei to be previously active, as our present evidence suggests. Drawing a simplistic scenario, we assumed the mass of the main body to be in the range of approximately $10^4 M_\odot$–$10^7 M_\odot$ for both eRO-QPE1 and eRO-QPE2 and computed the expected period decrease of a compact binary due to emission of gravitational waves. We inferred that a supermassive black hole binary with a mass ratio of the order of unity[23] is unlikely given the properties of the observed optical, ultraviolet (UV), infrared and X-ray emission in QPEs and the lack of evident periodicity and/or strong period decrease therein. If QPEs are triggered by the presence of a secondary orbiting body, our data suggest its mass ($M_2$) to be much smaller than the main body. This is in agreement with at least one proposed scenario for the origin of GSN 069, for which the average luminosity in a QPE cycle can be reproduced by a periodic mass-inflow rate from a white dwarf orbiting the black hole with a highly eccentric orbit[10]. Our current data for eRO-QPE1 only exclude $M_2$ larger than approximately $10^6 M_\odot$ (~$10^4 M_\odot$) for zero (high, ~0.9) eccentricity (as a function of the mass of the main body, Extended Data Fig. 7a); instead, for eRO-QPE2 we can already argue that only an orbiting $M_2$ lower than approximately $10^4 M_\odot$ (~$10 M_\odot$) is allowed for zero (~0.9) eccentricity (Extended Data Fig. 7b). More details are reported in Methods section 'On the presence of an orbiting body'.

Future X-ray observations on longer temporal baselines (months or years) will help to constrain or rule out this scenario and to monitor the possible orbital evolution of the system. This picture is also reminiscent of a suggested formation channel of extreme-mass-ratio inspirals[24,25] and it could make QPEs their electromagnetic messenger[13,26]. Regardless of their origin, the QPEs seen so far seem to be found in relatively

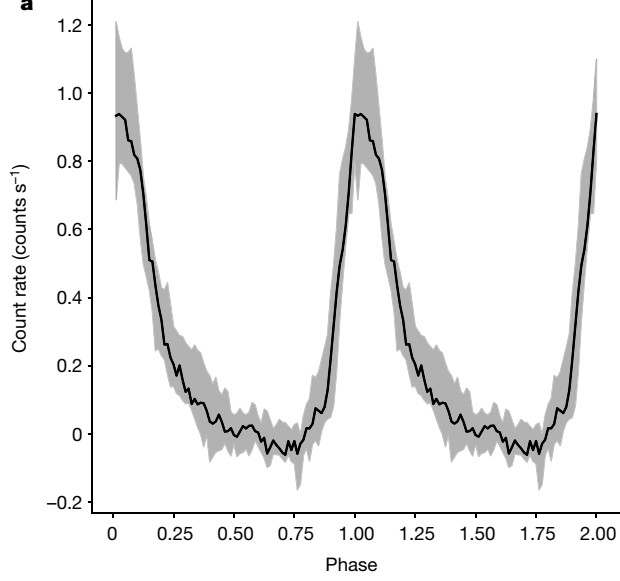

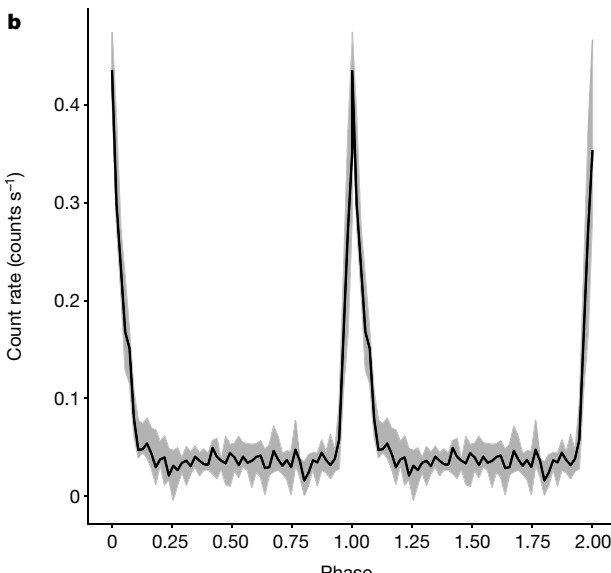

**Fig. 3 | Phase-folded light curves. a, b,** Median light curve profile (with related 16th and 84th percentile contours) for eRO-QPE1 (**a**) and eRO-QPE2 (**b**), folded at the eruption peaks (see Methods).

low-mass supermassive black holes ($\sim 10^5 M_\odot$–$10^7 M_\odot$) and finding more will help us to understand how black holes are activated in low-mass galaxies, which—although so far a poorly explored mass range in their co-evolution history[27,28]—is crucial for synergies with future Laser Interferometer Space Antenna (LISA) gravitational wave signals[29].

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

## Methods

### Blind search for QPEs with eROSITA

eROSITA[16] is the main instrument aboard the Spectrum-Roentgen-Gamma (SRG) mission (R. Sunyaev et al., manuscript in preparation), which was launched on 13 July 2019. On 13 December 2019 it started the first of eight all-sky surveys (eRASS1–eRASS8), each completed in six months, observing in the 0.2–8-keV band. In each survey, as the field of view moves every point of the sky is observed for ~40 s every ~4 h with the number of times (typically six) varying with the location in the sky, increasing towards high ecliptic latitudes. Our search for QPE candidates starts with a systematic screening of all eROSITA light curves, produced for each detected source on a weekly basis by the eROSITA Science Analysis Software (eSASS; H. Brunner et al., manuscript in preparation). Light curves are binned to yield one data point for each 4-h revolution (called an 'eROday'). A light curve generated by the eSASS pipeline will trigger a 'QPE alert' if it shows two or more high-count states with (at least) one low-count state in between (see Figs. 1a, 2a as examples) in any of its standard energy bands (0.2–0.6 keV, 0.6–2.3 keV, 2.3–5.0 keV). As thresholds, we fixed a relative count-rate ratio (including uncertainties) of 5 if both high and low states are detected, or 3 if the low-count state is consistent with the background. Since neither the survey scans nor QPEs are strictly periodic, every eRASS can be treated as an independent sky to find new candidates. This produces a census of X-ray sources varying on hour-long timescales for each eRASS, albeit only for the specific intermittent pattern described above. Unsurprisingly, the vast majority of the automatically generated alerts are produced by Galactic sources (mainly flaring coronally active stars), but we can filter them out by finding the multi-wavelength counterpart associated with every X-ray source (M. Salvato et al., manuscript in preparation). Good QPE candidates are then selected by screening the handful of alerts with a secure or possible extra-galactic counterpart. Thanks to this process, we identified the two best eROSITA QPE candidates that were worth immediate follow-up, promptly obtained with both XMM-Newton and, in one case, NICER. Given the success of our initial search over the first nine months of the survey, we are confident that we can detect up to around 3–4 good eROSITA QPE candidates every year. Therefore, by the end of the last eROSITA all-sky survey in December 2023 this search may provide a sample of up to approximately 10–15 QPEs.

**The two eROSITA QPEs.** The first QPE, here named eRO-QPE1, is eRASSU J023147.2-102010, located at the astrometrically corrected X-ray position of $RA_{J2000}$, $dec._{J2000} = (02:31:47.26, -10:20:10.31)$, with a total $1\sigma$ positional uncertainty of ~2.1″. It was observed ten times between 16 and 18 January 2020 during eRASS1 with 339 s of total exposure. Using the Bayesian cross-matching algorithm NWAY[30], we associated eRO-QPE1 with the galaxy 2MASS 02314715-1020112 at $RA_{J2000}$, $dec._{J2000} = (02:31:47.15, -10:20:11.22)$. The second QPE, here named eRO-QPE2, is eRASSU J023448.9-441931, located at the astrometrically corrected X-ray position of $RA_{J2000}$, $dec._{J2000} = (02:34:48.97, -44:19:31.65)$, with a total positional uncertainty of ~3.2″. It was observed 11 times between 23 and 24 June 2020 during eRASS2. It was associated via the same method[30] with 2MASX J02344872-4419325, a galaxy at $RA_{J2000}$, $dec._{J2000} = (02:34:48.69, -44:19:32.72)$. Both galaxies are in the DESI Legacy Imaging Surveys[31] DR8 footprint (Extended Data Figs. 1a, 2a). X-ray XMM-Newton positions were corrected with the 'eposcorr' task cross-correlating the sources in the X-ray image with external optical and infrared catalogues. The counterpart of the QPE itself was excluded from the cross-correlation to obtain a more unbiased estimate of the possible offset from the nucleus. The XMM-Newton X-ray positions are consistent with the nuclei of these galaxies. We took optical spectra of both galaxies with the Southern African Large Telescope (SALT)[32] and measured spectroscopic redshifts of 0.0505 and 0.0175 for eRO-QPE1 and eRO-QPE2, respectively (Extended Data

Figs. 1b, 2b). More details are shown in Methods sections 'Data reduction' and 'The host galaxies of the QPEs'.

**Previous X-ray activity.** eRO-QPE1 has not previously been detected in X-rays, although upper limits can be obtained from the XMM-Newton upper limits server for ROSAT[33], both from the survey and a pointed observation (taken in 1991 and 1992, with ~270 s and ~5,300 s, respectively), and the XMM-Newton Slew Survey[34] (taken in 2004, 2007, 2008 and 2017, all between ~3 s and ~8 s of exposure). The ROSAT pointed observation puts a stringent upper limit at $\leq 3.8 \times 10^{-14}$ cgs (cgs, erg s$^{-1}$ cm$^{-2}$) in the 0.2–2.0-keV band. However, given the very short exposures compared with the timescales of eRO-QPE1, we cannot rule out that QPEs were already ongoing and that all previous missions caught eRO-QPE1 in a faint state. As with eRO-QPE1, eRO-QPE2 has not been previously detected in X-rays. Upper limits were again computed for ROSAT (taken in 1990, ~480 s of exposure) and the XMM-Newton Slew survey (taken in 2004, 2008, 2012 and 2013, all between ~4 s and ~8 s). The most stringent upper limit, at $\leq 8.8 \times 10^{-14}$ cgs in the 0.2–2.0-keV band, comes from ROSAT. It is slightly below the flux observed by XMM-Newton in quiescence in the same band (Extended Data Fig. 4), perhaps indicating that the QPE behaviour only started more recently. For both QPE sources however, the ROSAT and Slew exposures are much shorter than the evolving timescales (the QPE quasi-period and its dispersion), hence they do not provide meaningful constraints on the start of the QPE behaviour.

### Data reduction

In this section we report details of the processing of the complete dataset. We show a summary of the observations in Extended Data Table 1.

**eROSITA.** Members of the German eROSITA consortium (eROSITA-DE) have full and immediate access to survey data at Galactic longitudes $180° < l < 360°$. These data were processed using eSASS v946 (H. Brunner et al., manuscript in preparation). For eRO-QPE1 (eRO-QPE2), photons were extracted choosing a circular aperture of radius 80″ (67″), and background counts were extracted from an annulus (off-centre circle) of inner and outer radii 178″ (382″) and 996″, respectively, excluding all the other sources detected within the area. eRO-QPE1 was detected with a detection likelihood of 440 and a total number of 119 counts in the 0.2–5.0-keV band. eRO-QPE2 was detected with a detection likelihood of 125 and a total number of 48 counts in the 0.2–5.0-keV band.

**XMM-Newton.** XMM-Newton data from EPIC MOS1, MOS2[35] and EPIC-pn[36] cameras and the Optical Monitor (OM)[37] were processed using standard tools (SAS v.18.0.0 and HEAsoft v.6.25) and procedures. Event files from EPIC cameras were filtered for flaring particle background. Source (background) regions were extracted within a circle of 38″ and 34″ in eRO-QPE1 and eRO-QPE2, respectively, centred on the source (in a source-free region). eRO-QPE1 was consecutively observed three times with the U filter, then seven times with UVW1 and nine (eight) times with the UVM2 in the first (second) XMM-Newton observation, each exposure ~4,400-s long. The source was detected only in the U and UVW1 with mean magnitudes ~19.9 and ~20.3 in both XMM-Newton observations (OM light curves in Fig. 1c). eRO-QPE2 was consecutively observed twice with the U filter, then ten times with UVW1, six with UVM2 and three with UVW2; all exposures were 4,400 s. It was almost always detected in all filters with mean magnitudes of ~17.4, ~17.5, ~18.0 and ~18.1, for U, UVW1, UVM2 and UVW2 filters, respectively (OM light curves in Fig. 2c). eRO-QPE2 was flagged as extended in the U, UVW1 and UVM2 filters, and therefore the reported absolute magnitudes include at least some contamination from the host galaxy.

**NICER.** NICER's X-ray Timing Instrument (XTI)[38,39] onboard the ISS observed eRO-QPE1 between 17 August 2020 and 31 August 2020. Beginning late on 19 August, high-cadence observations were performed during almost every ISS orbit, which is roughly 93 min. All the data were

processed using the standard NICER Data Analysis Software (NICER-DAS) task 'nicerl2'. Good time intervals (GTIs) were chosen with standard defaults, yielding ~186 ks of exposure time. We further divided the GTIs into intervals of 128 s, and on this basis we extracted the spectra and applied the '3C50' model (R.R. et al., submitted) to determine the background spectra. The light curve for eRO-QPE1 in soft X-rays (Fig. 1d) was determined by integrating the background-subtracted spectrum for each 128-s GTI over the range 0.3–1.0 keV. More detailed spectral analyses of these data will be discussed in a follow-up paper.

**SALT.** Optical spectra of eRO-QPE1 and eRO-QPE2 were obtained using the Robert Stobie Spectrograph (RSS)[40] on the Southern African Large Telescope (SALT)[32] in September 2020 on the nights of the 24th and the 8th, respectively. The PG900 VPH grating was used to obtain pairs of exposures (900 s and 500 s, respectively) at different grating angles, allowing for a total wavelength coverage of 3,500–7,400 Å. The spectra were reduced using the PySALT package, a PyRAF-based software package for SALT data reductions[41], which includes gain and amplifier cross-talk corrections, bias subtraction, amplifier mosaicing, and cosmetic corrections. The individual spectra were then extracted using standard Image Reduction and Analysis Facility (IRAF) procedures, wavelength calibration (with a calibration lamp exposure taken immediately after the science spectra), background subtraction and extraction of one-dimensional spectra. We could only obtain relative flux calibrations, from observing spectrophotometric standards in twilight, owing to the SALT design, which has a time-varying, asymmetric and underfilled entrance pupil[42].

### X-ray spectral analysis
In this work, X-ray spectral analysis was performed using v3.4.2 of the Bayesian X-ray Analysis software (BXA)[43], which connects a nested sampling algorithm (UltraNest[44]; J.B., manuscript in preparation) with a fitting environment. For the latter, we used XSPEC v12.10.1[45] with its Python-oriented interface pyXSPEC. eROSITA source-plus-background spectra were fitted including a model component for the background, which was determined via a principal component analysis from a large sample of eROSITA background spectra[46] (J.B. et al., manuscript in preparation). XMM-Newton EPIC-pn spectra were instead fitted using wstat, namely the XSPEC implementation of the Cash statistic[47], given the good count statistics in both source and background spectra. We quote, unless otherwise stated, median values with the related 16th and 84th percentiles and upper limits at 1$\sigma$. Results are also reported in Extended Data Tables 2, 3.

**eRO-QPE1.** For eRO-QPE1, both eROSITA and XMM-Newton EPIC-pn spectra were fitted with a simple absorbed black body (using the models tbabs[48] and zbbody) or accretion disk (tbabs and diskbb[49]), with absorption frozen at the Galactic equivalent hydrogen column density ($N_H$) of $N_H \approx 2.23 \times 10^{20}$ cm$^{-2}$, as reported by the HI4PI Collaboration[50]. For eROSITA, we jointly extracted and analysed spectra of the faint states (red points in Fig. 1a) and, separately, of the two bright states observed in eRASS1 (orange points in Fig. 1a). In the eROSITA bright states the temperature, in terms of $k_B T$ ($k_B$, Boltzmann's constant; $T$, temperature) in eV, is $138^{146}_{131}$ eV and $180^{195}_{168}$ eV, using zbbody and diskbb as source models, respectively. The related unabsorbed rest-frame 0.5–2.0-keV fluxes are $1.6^{1.8}_{1.4} \times 10^{-12}$ cgs and $1.5^{1.7}_{1.4} \times 10^{-12}$ cgs, respectively. The eROSITA spectrum of the faint states combined is consistent with background, with the temperature and unabsorbed rest-frame 0.5–2.0-keV flux constrained to be ≤124 eV (≤160 eV) and ≤3.5×10$^{-14}$ cgs (≤3.4×10$^{-14}$ cgs) for zbbody (diskbb). We also analysed the observations of eRO-QPE1 obtained six months later during eRASS2, which triggered our QPE search again: two bright states were observed separated by several faint ones, with fluxes consistent with eRASS1.

We performed time-resolved X-ray spectral analysis on XMM-Newton data, extracting a spectrum in each 500-s time bin of the EPIC-pn light curve, with the exception of the quiescence spectrum, which was extracted and analysed combining all the related time bins of both observations (that is, before $t \approx 26,500$ s in eRO-QPE1-XMM1 and before $t \approx 35,788$ s in eRO-QPE1-XMM2, with times as defined in Fig. 1c). Fit results obtained using XMM-Newton EPIC-pn spectra with diskbb as the source model component are shown in Extended Data Fig. 3. Furthermore, we show for visualization three EPIC-pn spectra and related best-fit models (Extended Data Fig. 5a) corresponding to the quiescence phase and the peak of both XMM-Newton observations. A more thorough X-ray spectral analysis with other models and additional components for the bright phase will be presented in future work.

**eRO-QPE2.** For eRO-QPE2, eROSITA's faint and bright phases were also separately combined and analysed (Fig. 2a, b). The faint state as observed by eROSITA is consistent with background. The temperature and normalization of the source cannot be constrained, thus we only quote an upper limit for the unabsorbed rest-frame 0.5–2.0-keV flux of ≤1.9×10$^{-14}$ cgs (≤5.7×10$^{-14}$ cgs) using zbbody (diskbb). The spectrum of the eROSITA bright states constrains the temperature to $164^{182}_{149}$ eV and at $209^{241}_{185}$ eV, using zbbody and diskbb as source models, respectively. The related unabsorbed rest-frame 0.5–2.0-keV fluxes are $1.4^{1.8}_{1.2} \times 10^{-12}$ cgs and $1.5^{1.8}_{1.2} \times 10^{-12}$ cgs, respectively. The triggering eROSITA observation was obtained during eRASS2, although a single bright state (thus not satisfying our trigger criterion) was also detected in eRASS1 with the same flux level. For eRO-QPE2, in addition to the Galactic column density ($N_H \approx 1.66 \times 10^{20}$ cm$^{-2}$)[50] we included an absorption component at the redshift of the host galaxy (that is, with the models tbabs, ztbabs, and zbbody or diskbb). This excess absorption was inferred to be present on the basis of the XMM-Newton spectrum (see below).

For XMM-Newton, we performed time-resolved X-ray spectral analysis for each 150-s time bin of the EPIC-pn light curve. The absorption, in addition to the Galactic value, was first fitted in the XMM-Newton quiescence spectrum, which was extracted combining all the low states in the XMM-Newton light curve (Fig. 2c, Extended Data Fig. 4). The fit yielded $N_H = 0.35^{0.40}_{0.30} \times 10^{22}$ cm$^{-2}$. In all other observations, including all eROSITA spectra and the rises, peaks and decays in the XMM-Newton light curve, the additional $N_H$ was left free to vary between the 10th and 90th percentile of the fitted posterior distribution of the quiescent spectrum. Under the assumption that absorption did not vary throughout the observation, this ensures that no spurious effects are imprinted on the fit temperature and normalizations owing to degeneracies with $N_H$; at the same time, in this way parameters are marginalized over a reasonable range in $N_H$. Freezing the value instead would artificially narrow the uncertainties on the temperature and normalizations. Fit results obtained with diskbb as the source model are shown in Extended Data Fig. 4. Furthermore, we show for visualization the EPIC-pn spectra and best-fit models of the quiescence and peak phases (Extended Data Fig. 5b). Similar results are obtained using zbbody as the source model.

### Timing analysis
In Fig. 3 we show the median (with related 16th and 84th percentile contours) light-curve profiles obtained by folding the light curve at the eruption peaks. First, a random representative burst is selected and cross-correlated with the whole light curve. The peaks of this cross-correlation identify the times when the phase is zero. Data are then folded at these phase-zero times to obtain a template median profile, which is then used to repeat the same operation and yield Fig. 3. A phase bin of ~0.1 corresponds to ~6,600 s and ~820 s for eRO-QPE1 and eRO-QPE2, respectively. Moreover, XMM-Newton and NICER light curve profiles were fitted with UltraNest[44]. Motivated by the strong asymmetry in eRO-QPE1 (Figs. 1c, d, 3a), we adopted a model with Gaussian rise and an exponential decay, a generic model often adopted for transients[51]. eRO-QPE2, on the other hand, can be fitted with a simple Gaussian profile (Fig. 3b), possibly owing to the much

shorter timescales. Here we simply highlight the most evident results of timing analysis; a more in depth study of the variability properties of QPEs is deferred to future work. Here the modelling allows us to determine mean values for the duration and recurrence time of the QPEs, which were used for comparison with models of accretion instabilities (see Methods section 'On accretion flow instabilities') and compact object binaries (see Methods section 'On the presence of an orbiting body'). The mean rise-to-decay duration for eRO-QPE1, as observed from the NICER light curve (Fig. 1d), is ~7.6 h (dispersion of ~1.0 h), and the mean peak-to-peak separation is ~18.5 h (dispersion of ~2.7 h). The related duty cycle (here computed simply as mean duration over mean separation) is ~41%. Conversely, eRO-QPE2 shows much narrower and more frequent eruptions (see Fig. 2c): the mean rise-to-decay duration is ~27 min (dispersion of ~3 min), with a mean peak-to-peak separation of ~2.4 h (dispersion of ~5 min) and a duty-cycle of ~19%.

### The host galaxies of the QPEs

Very little was known on both galaxies from published multi-wavelength catalogues, except for WISE infrared monitoring, indicating W1 − W2 emission consistent with zero, which is typical of inactive galactic nuclei, for the last few years. Most of our knowledge is based on optical spectra taken with SALT after the X-ray observation. The optical counterpart of eRO-QPE1 is classified as a passive galaxy from the absence of emission lines (Extended Data Fig. 1b), whereas eRO-QPE2 shows very strong and narrow [O II], Hβ, [O III], Hα, [N II] and [S II] in emission (Extended Data Fig. 2b). The high [O II]/[O III] value and that Hβ is as strong as [O III] are already strongly indicative that star-forming processes are the dominant ionization mechanism[52]. We computed the flux ratios $\log([\text{O III}]/\text{H}\beta) = -0.05$, $\log([\text{O II}]/\text{H}\beta) = 0.44$ and $\log([\text{N II}]/\text{H}\alpha) = -0.68$, as well as the [O II] equivalent width (EW) $\log \text{EW}_{[\text{OII}]} = 2.56$ and $D_{4,000} = 1.26$, where $D_{4,000}$ is the ratio of the continuum level after and before the 4,000 Å break[53]. Using standard line diagnostic plots[54] we can confirm that indeed eRO-QPE2 can be classified as star-forming. Spectroscopic classification of future QPEs will be crucial to confirm whether their host galaxies are indeed preferentially inactive, as our pilot study suggests, or not. A first census in a statistically significant sample may bring new insights, as has been the case for other transients such as tidal disruption events (TDEs)[55–58].

A preliminary analysis of the properties of the host galaxies of the QPEs was performed by fitting the optical spectra (Extended Data Figs. 1b, 2b) with Firefly[59,60]. We first re-normalized the flux of the optical spectra using the most recent g-band and r-band archival magnitudes, because SALT spectra are not calibrated to absolute values[42]. For eRO-QPE1, gri-band photometry ($g = 18.7 \pm 0.06$ mag, $r = 18.0 \pm 0.05$ mag, $i = 17.8 \pm 0.05$ mag) was taken on 30 July 2020 with the Rapid Response Robotic Telescope at Fan Mountain Observatory, indicating that the source did not change substantially with respect to archival photometry[61]. The total stellar masses inferred with Firefly from the optical spectra are $M_* \approx 3.8^{+0.4}_{-1.9} \times 10^9 M_\odot$ and $1.01^{+0.01}_{-0.50} \times 10^9 M_\odot$ for eRO-QPE1 and eRO-QPE2, respectively. Systematic errors and degeneracy due to the use of different stellar population models[62] would push $M_*$ to higher values for eRO-QPE1 (~$4.8 \times 10^9 M_\odot$) and lower values for eRO-QPE2 (~$0.6 \times 10^9 M_\odot$), enhancing their relative difference. Firefly also yields an estimate of the age of the stellar population and the star-formation rate (SFR), although for medium and low signal-to-noise ratios these estimates are more prone to biases[59]. For eRO-QPE2, the mean signal-to-noise ratio (~23) is high enough to yield a fairly reliable $\text{SFR} \approx 0.078^{+0.001}_{-0.066} M_\odot$ yr$^{-1}$, which is also consistent within uncertainties with the SFR that can be estimated from the [O II] and Hα luminosities[63,64]. For eRO-QPE1, the mean signal-to-noise ratio (~8) is lower and no reliable estimate of the SFR was obtained. We therefore inferred an upper limit of ~$0.01 M_\odot$ yr$^{-1}$ from the absence of narrow emission lines[63,64]. We report in Extended Data Fig. 6 the $M_*$–SFR plane with the two QPEs reported here, together with normal galaxies, and hosts of known TDEs[57] and changing-look AGN (CLAGN)[65], all taken below $z < 0.1$ and

within the Sloan Digital Sky Survey MPA-JHU DR7 catalogue[66]. Evidence is mounting that both TDEs[57,67] and CLAGN[65] might be over-represented in galaxies in the so-called 'green valley', perhaps indicating that they are activated in specific periods of galaxy co-evolution with their central black holes. For QPEs, a statistically meaningful sample still needs to be built before reaching any conclusion.

We have estimated that the host galaxy of eRO-QPE1 is more massive than that of eRO-QPE2. We here refrain from quoting absolute values for black hole masses using their scaling relations with the host galaxy properties, because our stellar masses are lower than those used to calibrate them[68]. However, it is worth mentioning that the relative ratio of ~4–8 in stellar masses between the galaxies of eRO-QPE1 and eRO-QPE2 would propagate to a black hole mass ratio of the order of approximately 10 (ref. [68]). This is in line with the X-ray timing properties in eRO-QPE1 and eRO-QPE2, because their peak-to-peak separation and rise-to-decay duration scale roughly by the same amount. Finally, X-ray emission from eRO-QPE1 and eRO-QPE2 was observed to be positionally consistent with the galaxy nucleus for both objects (Extended Data Figs. 1a, 2a; Methods section 'The two eROSITA QPEs'). If a future QPE is found in a more nearby galaxy we can aim to constrain more precisely the X-ray position with respect to the galactic nucleus. This will allow us to determine conclusively whether or not these phenomena are nuclear.

### On accretion flow instabilities

Accretion disks[69] with an accretion rate such that radiation pressure dominates in the inner flow are thought to be subject to thermal–viscous instabilities[70]. The net result of these instabilities is that the luminosity is predicted to oscillate[5–8] with timescales and amplitude proportional to the black hole mass and bolometric luminosity[71,72]. The predicted light curves profiles show first a stable and slow rise in luminosity, as both temperature and surface density increase while matter is slowly accumulated. Thereafter a sharp luminosity burst is produced by a runaway increase (decrease) in temperature (surface density) propagating outwards within the unstable region. Finally, the inner flow, devoid of the matter supply, cools down rapidly and cycles back to the initial stable state with low temperature and density. Both heating and cooling fronts propagate following thermal timescales[6], where $\tau_{\text{th}} \approx \alpha^{-1}(GM_{\text{BH}}/R^3)^{-1/2}$ (where $R$ is the distance from the black hole). These so-called limit-cycle or 'heartbeat' instabilities have been successfully applied to a few accreting sources across all mass scales, for instance to the stellar-mass black holes GRS 1915+105[20,21,73], IGR J17091-3624[22] and 4XMM J111816.0-324910[74] and to supermassive black holes in a statistical fashion[71,72]. The similarity of their timing properties with those of QPEs in GSN 069 is remarkable and naturally led to the proposed connection with limit-cycle instabilities for that object. In particular, the symmetry of the eruptions in GSN 069 was compared to the fast heating and cooling phases of the instability[1], both following similar $\tau_{\text{th}}$ under the assumption of invariant $\alpha$ across the two phases[75]. The lack of a slow rise before the eruptions in QPEs, predicted by the instability models, could be due to our limited coverage of the full disk temperature profile in the soft X-ray band.

With the observation of the two QPEs we report here we can now argue against at least this type of accretion disk instability as the origin of QPEs. Specifically, the strong asymmetric nature of the eruptions in eRO-QPE1, which show a faster rise and a much slower decay (Fig. 3a), argues against this interpretation. Qualitatively, our data would suggest that QPEs are not related to $\tau_{\text{th}}$, because $\alpha$ is not expected to change between the hot and cold phases in AGN[75]. Moreover, if instead it is the front propagation timescales, following $\tau_{\text{front}} \approx (H/R)^{-1} \tau_{\text{th}}$ (where $H$ is the vertical scale height of the disk), or viscous timescales, following $\tau_{\text{visc}} \approx (H/R)^{-2} \tau_{\text{th}}$, that regulates the rise (decay) in the cold (hot) phase, it would imply a thicker flow in the cold and stable phase than in the hot and unstable phase. This runs contrary to the theoretical expectation that unstable flows should be thicker[6]. The limit-cycle oscillation theory further predicts that once the period, duty cycle

and luminosity amplitude are known and a viscosity prescription for the accretion flow is adopted, there are only specific values of $M_{BH}$ and $\alpha$ that unstable sources are allowed to follow[8]. Here we adopt for eRO-QPE1 (eRO-QPE2) a peak-to-peak period $T_{pp} = 18.5$ h (2.4 h), an amplitude $A \approx 294$ (11) and a duty cycle $D = 41\%$ (19%). The amplitude $A$ is the ratio of the disk luminosity (computed within the 0.001–100-keV range) for peak and quiescence, taken as proxy of the maximum and minimum bolometric luminosity, and $D$ is here defined as the ratio of the flare duration (rise-to-decay $T_{rd}$) and the period $T_{pp}$. We begin by adopting a standard viscosity prescription, with the average stress between the annuli proportional to the gas plus radiation pressure[69] $P_{tot}$. The allowed $M_{BH}$ and $\alpha$ values for eRO-QPE1 (eRO-QPE2) are $M_{BH} \approx 4 \times 10^6 M_\odot$ and $\alpha \approx 5$ ($M_{BH} \approx 3 \times 10^6 M_\odot$ and $\alpha \approx 3$), therefore an unphysically high viscosity parameter would be required. Considering alternative viscosity prescriptions[5,8], for eRO-QPE1 (eRO-QPE2) a more reasonable $\alpha \approx 0.1$ or 0.01 would correspond to allowed $M_{BH} \approx 2.4 \times 10^3 M_\odot$ or $M_{BH} \approx 60 M_\odot$ ($M_{BH} \approx 4.3 \times 10^3 M_\odot$ or $M_{BH} \approx 30 M_\odot$), respectively. The above combinations are either unphysical or very unlikely. Adopting $\alpha \approx 0.2$ and alternative viscosity prescriptions, eRO-QPE1 (eRO-QPE2) would yield an intermediate-mass black hole (IMBH) at $M_{BH} \approx 0.9 \times 10^4 M_\odot$ ($M_{BH} \approx 1.6 \times 10^4 M_\odot$) accreting at ~0.1 (~0.3) of the Eddington limit in quiescence and at ~30 (~3) times the Eddington limit at the peak. However, this IMBH scenario would not account for the opposite asymmetry shown by eRO-QPE1 compared to the theoretical predictions, nor would the resulting thermal timescales be self-consistent for either of the two: for eRO-QPE1 (eRO-QPE2) $\tau_{th} \approx 20$ s (35 s) at $20r_g$ adopting $M_{BH} \approx 0.9 \times 10^4 M_\odot$ ($M_{BH} \approx 1.6 \times 10^4 M_\odot$), which is orders of magnitude smaller than the observed QPE durations, and the rise-to-peak times would be only reconciled with $\tau_{th}$ at ~$780r_g$ (~$250r_g$). Analogous results can be obtained using the observed properties of RX J1301.9+2747[2], adopting $T_{pp} \approx 20$ ks (or the second period $T_{pp} \approx 13$ ks), $D = 6\%$ (9%) and $A \approx 9.4$, the latter obtained taking the ratio of the quoted quiescence and peak 0.3–2.0-keV flux as proxy for a bolometric luminosity ratio: adopting[2] $\alpha \approx 0.15$ the allowed black hole mass is ~$2.2 \times 10^4 M_\odot$ (~$1.5 \times 10^4 M_\odot$), much lower than the quoted[2,76] ~$(0.8$–$2.8) \times 10^6 M_\odot$.

When a given source is in a 'sweet-spot' regime in mass accretion rate, some more recent modified viscosity prescriptions of accretion disks predict the presence of a narrow unstable zone placed within an inner inefficient advection-dominated flow and an outer standard geometrically thin and stable flow[9]. This model would reduce the propagation timescales by a factor of approximately d$R$/$R$, where d$R$ is the radial extent of the unstable zone at a distance $R$ from the black hole, which may help reconcile the model with the dramatic and fast variability observed in CLAGN[77]. This would not, however, change the inconsistency with the asymmetric shape of the QPEs we report here, nor was it successful in modelling all the observables in GSN 069[9]. In summary, our data for both of the QPEs reported here are inconsistent with published models for radiation pressure instability[5–9]. The role of more complex or exotic phenomenology[9] should be further explored.

We also note that a fast-rise exponential decay profile—such as the one in eRO-QPE1—can be naturally produced by ionization instability models, which are used for some bursting stellar-mass accreting compact objects[78]. To our knowledge there is no evidence so far of such instabilities taking place in AGN[79]. In addition, the predicted timescales are many orders of magnitude longer than QPEs for both AGN[79–81] and IMBH masses[82].

Finally, we discuss disk warping and tearing induced by Lense–Thirring precession[83,84], which has been recently qualitatively compared also to QPE sources[85]. In this work we presented evidence of QPEs being observed in previously inactive galaxies, therefore the accretion flow in these systems should be young. Moreover, a key element of disk warping and tearing due to Lense–Thirring precession is that mass needs to flow in from large inclination with respect to the black hole spin. Both conditions are satisfied if the accretion flow is formed, for instance, by a fully stripped TDE. However, in this case the warped

inner flow would be damped very fast[86], which would be in contrast with QPEs lasting at least months[1] (Figs. 1, 2) or even years[2]. A more quantitative comparison is beyond the reach of this work and of current disk warping and tearing simulations, but this is a promising scenario worth exploring in the future.

## On the presence of an orbiting body

Periodic variability is also often associated with binary systems of compact objects[23] and the connection with the quasi-periodic nuclear emission observed in QPEs of interest for future work. We here assume the main body to be a supermassive black hole ranging between approximately $10^4 M_\odot$ and $10^7 M_\odot$ and we first consider the presence of a second orbiting supermassive black hole with a similar mass. There are several reasons which, when combined, disfavour such a scenario. First, simulations show that the accretion flow of such objects is composed by a circum-binary disk with two inner small mini-disks[87–89], which are thought to produce a quasi-sinusoidal modulated emission[90,91]. This signature can be detected in transient surveys[92,93] or in single sources[94], with a well known extreme case being OJ 287[95,96]. However, so far there is no evidence of such variability in optical and UV data[1] of QPEs (Fig. 1c, 2c), in particular in eRO-QPE1, which was covered in the $g$ and $r$ bands by the Zwicky Transient Facility DR3[97] until the end of 2019. Nor can this prediction be reconciled with the dramatic non-sinusoidal eruptions observed in X-rays, even in the case of binary self-lensing[98] which can produce sharper bursts, albeit achromatic, and therefore in contrast with the energy dependence of QPEs[1,2]. Moreover, we do not observe peculiar single- or double-peaked emission lines[99–101] and this cannot be reconciled by enhanced obscuration[102], because infrared photometry in QPEs is not AGN-like (WISE has observed stable W1 − W2 ≈ 0 emission for the past 6–7 yr) and X-rays do not indicate the presence of strong absorption. Second, supermassive black hole binaries are expected to form mostly via galactic mergers[103,104], but the host galaxies of the two QPEs we report here look unperturbed (Extended Data Figs. 1a, 2a). Perhaps most importantly, a binary of supermassive black holes observed with a periodicity of the order of hours, such as the four observed QPEs, would show a large period derivative due to gravitational wave emission, and would be relatively close to merger. To have (at least) four such objects very close to merger within $z \approx 0.02$–$0.05$ is very unlikely[105] and would imply that they are much more common in the local Universe than observations suggest[92,93].

Under the simplified assumption that the orbital evolution is dominated by gravitational waves emission, Extended Data Fig. 7 shows the allowed parameter space in terms of the derivative of the period, $\dot{P}$, and $M_2$ for a range of $M_{BH,1} \approx 10^4 M_\odot$–$10^7 M_\odot$ and zero- or high-eccentricity orbit ($e_O \approx 0.9$), given the rest-frame period of both eRO-QPE1 and eRO-QPE2. We have additionally imposed $M_2 \leq M_{BH,1}$. For both sources we can draw a tentative line at the minimum period derivative that, if present, we would have measured already within the available observations: somewhat conservatively, we adopt a period decrease of one cycle over the 15 observed by NICER for eRO-QPE1 and the nine observed by XMM-Newton for eRO-QPE2 (Figs. 1d, 2c). Our constraint on $\dot{P}$ is not very stringent for eRO-QPE1 and only high $M_2$ and eccentricities are disfavoured; instead, for eRO-QPE2 only an orbiting IMBH, or smaller, is allowed for zero eccentricity, whereas only a stellar-mass compact object is allowed for high eccentricity ($e_O \approx 0.9$). Future observations of eRO-QPE1 and eRO-QPE2 in the next months may lead to tighter constraints on the mass and eccentricity of the putative orbiting body.

The preliminary conclusion of our analysis is that, if QPEs are driven by the presence of an orbiting body around a central black hole, it is more likely that this is a compact object with a mass considerably smaller than the ~$10^4 M_\odot$–$10^7 M_\odot$ assumed for the main body. This scenario could make QPEs a viable candidate for the electromagnetic counterparts of the so-called extreme-mass-ratio inspirals (EMRIs)[11–13], with considerable implications for multi-messenger astrophysics and cosmology[14,15]. Interestingly, it has been recently suggested for GSN 069

that a stellar-mass compact object orbiting around a supermassive black hole could be the origin of QPEs: a white dwarf of ~$0.2M_\odot$ on a highly eccentric orbit ($e_O \approx 0.94$) could reproduce the mass inflow rate needed to produce the observed X-ray luminosity averaged over a QPE cycle[10]. This is reminiscent of a suggested, albeit still observationally elusive, EMRI-formation channel[13,24–26]. For GSN 069, a possible explanation of the QPE-free X-ray bright and decaying phase could be given by an accretion flow expanding and intercepting the body at a later time[1]; or if the orbiting body was originally a massive star and the stripped envelope produced the TDE-like behaviour of the past decade[1] while the remaining core started interacting with the newly born or expanded accretion flow only at a later stage, which would also explain the relatively small mass required by the white dwarf calculations[10]. For the other QPEs that did not show evidence of a past X-ray bright and decaying phase, this scenario is not necessary and the interaction with a second stellar-mass (or more massive) compact object could qualitatively reproduce the periodic behaviour (Extended Data Fig. 7b). Future X-ray observations of the known QPEs would help in further constraining the possible orbital evolution. It should be pointed out that these calculations so far only match the average observed QPE luminosity with the mass inflow rate required to produce it[10], but details on the exact physical mechanism that would produce these X-ray bursts are still unknown (see Methods section 'On accretion flow instabilities').

Finally, we address the lack of UV and optical variability in the scenario of an orbiting body. The X-ray plateau at minimum shows an almost featureless accretion disk thermal spectrum[1,2] (Extended Data Fig. 5), which could have been built up during the first orbiting cycles. This accretion flow should be unusually small due to the lack of a broad line region[3,4] (Extended Data Figs. 1b, 2b), which would respond in light-days and that, if present, should have been observed in the SALT spectra taken months after the X-ray QPEs. The lack of strong UV and optical variability might be then due to the fact that the accretion disk is not large enough to even emit strong enough UV–optical radiation to emerge above the galaxy emission, which we can assume to be most of the observed $L \approx 4.0 \times 10^{41}$ erg s$^{-1}$ ($L \approx 4.3 \times 10^{41}$ erg s$^{-1}$) in the OM-UVW1 filter at 2,910 Å for eRO-QPE1 (eRO-QPE2). Using a simplified but physically motivated accretion disk model[106] for a spin-zero black hole accreting at ~0.1 of the Eddington limit, we computed the distance at which the bulk of 2,910-Å radiation would be emitted, namely ~1,100$r_g$ and ~500$r_g$ for masses of $10^5 M_\odot$ and $10^6 M_\odot$, respectively. This would shift to even larger radii for increasing accretion rate (for example, ~1,850$r_g$ and ~860$r_g$ at ~0.5 of the Eddington limit), whereas even for high spinning sources (dimensionless spin parameter ~0.9) the peak of OM-UVW1 flux would still come from ~775$r_g$ and ~360$r_g$. Furthermore, the predicted OM-UVW1 disk luminosity would be at least one or two orders of magnitude lower than the observed $L \approx 4.0 \times 10^{41}$ erg s$^{-1}$ in the most luminous scenario. Therefore, even a UV–optical eruption 100 times brighter than the plateau would be barely detectable above the galaxy component.

**Predicted numbers.** Detailed self-consistency calculations on the predicted rate of such EMRI events, as compared to QPE rates, are required but are beyond the scope of this paper. Instead, we can provide here a rough model-independent estimate of the expected number of observed QPEs, regardless of their origin. Convolving the black hole mass function[107] between approximately $10^{4.5}M_\odot$ and $10^{6.5}M_\odot$ up to $z \approx 0.03$ with the eROSITA sensitivity yields approximately 100 sources, a number that is then reduced with some educated guesses on some unknowns: during what fraction of their X-ray bright phase such sources undergo QPE behaviour (the biggest unknown; for example, >20% for GSN 069); how many such sources are obscured and missed (~2/3); how many times we detect ongoing QPEs given the eROSITA sampling (depends on the period and the burst duration; for example, ~20% for GSN 069). This results in a (extremely uncertain) number of the order of unity per eRASS scan in the eROSITA-DE hemisphere; however, this is in agreement with our pilot study of the first few months of eROSITA operations. Thus, the low observed numbers do not necessarily imply that these events are a rare phenomenon intrinsically and they can in fact be a fairly common product of the co-evolution of black holes with their host galaxies[28]. With a statistically meaningful sample of QPEs, inverting this calculation may allow us to constrain the black hole mass function in a poorly known mass regime[27].

## Data availability

With the exception of eROSITA proprietary data, the data used in this work are public and available from the corresponding data archives (XMM-Newton, http://nxsa.esac.esa.int/nxsa-web/#search; NICER, https://heasarc.gsfc.nasa.gov/docs/nicer/nicer_archive.html) or they will be soon, after the remaining proprietary period expires. Most of data may be available from the corresponding author on reasonable request.

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

**Acknowledgements** We are grateful to the XMM-Newton Project Scientist N. Schartel for the DDT observation of eRO-QPE2; we also thank the XMM-Newton Science Operations team for performing it along with the two approved ToOs on eRO-QPE1 with a relatively short notice before the end of the visibility window. R.A. thanks M. Gilfanov, K. Dennerl and S. Carpano for discussions. G.P. acknowledges funding from the European Research Council (ERC) under the European Union's Horizon 2020 research and innovation programme (grant agreement no. 865637). This work is based on data from eROSITA, the primary instrument aboard SRG, a joint Russian–German science mission supported by the Russian Space Agency (Roskosmos), in the interests of the Russian Academy of Sciences represented by its Space Research Institute (IKI), and the Deutsches Zentrum für Luft- und Raumfahrt (DLR). The SRG spacecraft was built by Lavochkin Association (NPOL) and its subcontractors, and is operated by NPOL with support from the Max Planck Institute for Extraterrestrial Physics (MPE). The development and construction of the eROSITA X-ray instrument was led by MPE, with contributions from the Dr. Karl Remeis Observatory Bamberg & ECAP (FAU Erlangen-Nuernberg), the University of Hamburg Observatory, the Leibniz Institute for Astrophysics Potsdam, and the Institute for Astronomy and Astrophysics of the University of Tübingen, with the support of DLR and the Max Planck Society. The Argelander Institute for Astronomy of the University of Bonn and the Ludwig Maximilians Universität München also participated in the science preparation for eROSITA. The eROSITA data shown here were processed using the eSASS/NRTA software system developed by the German eROSITA consortium. Some of the observations were obtained using the Southern African Large Telescope (SALT) as part of the Large Science Programme on transient (2018-2-LSP-001; PI: D.A.H.B.). Polish participation in SALT is funded by grant no. MNiSW DIR/WK/2016/07. D.A.H.B. is supported by the National Research Foundation (NRF) of South Africa. M.G. is supported by the Polish NCN MAESTRO grant 2014/14/A/ST9/00121. M.K. is supported by DFG grant KR 3338/4-1. This work is partly supported by the Optical and Near-infrared Astronomy Inter-University Cooperation Program and the Grants-in-Aid of the Ministry of Education of Japan.

**Author contributions** R.A. wrote the article, developed the method to search for QPE candidates, was PI of XMM-Newton and NICER follow-up observations, and performed most of the data analysis. A. Merloni was PS/PI of eROSITA and followed and improved the project, the strategy of XMM-Newton and NICER follow-up observing campaigns and the writing of the article. K.N. followed and improved the project, the strategy of XMM-Newton and NICER follow-up proposals and the writing of the article. J.B. developed part of the code used for spectral and timing analysis and helped R.A. with XMM-Newton data analysis and with the development of the method to search for QPE candidates. M. Salvato led the counterpart association and significantly contributed to the interpretation and analysis of the photometry and optical spectroscopy. D.P. and R.R. performed most of the NICER data analysis. J.C. performed some of the analysis of optical spectroscopy and improved its interpretation. G.L. performed the astrometry of eROSITA and XMM-Newton data. G.P. significantly improved the writing of the article and encouraged the pursuit of this project. A. Malyali and J.W. directly contributed to the analysis of XMM-Newton light curves, optical images and counterpart association. Z.A., K.G. and C.M. scheduled, monitored and processed NICER observations. D.A.H.B. instigated the SALT observations and M.G. processed related data. E.K. and M.K. improved the scientific interpretation and presentation of the manuscript. M.E.R.-C. contributed to development of the method to search for QPE candidates. A.R. contributed to XMM-Newton observing proposals and with A.S. led the triggering of SALT observations from the eROSITA-DE Consortium. M. Schramm performed and processed photometry data from the Rapid Response Robotic Telescope. All authors contributed to improve the projects from different sides throughout the process.

**Funding** Open access funding provided by Max Planck Society.

**Competing interests** The authors declare no competing interests.

**Additional information**
**Correspondence and requests for materials** should be addressed to R.A.

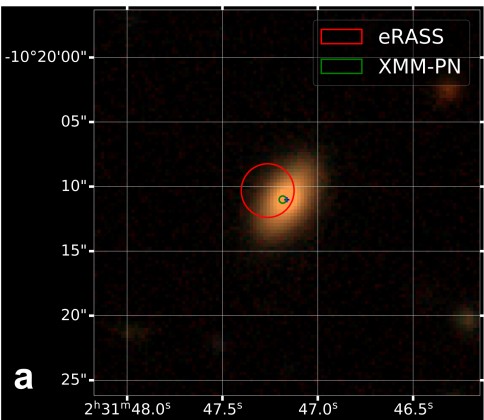

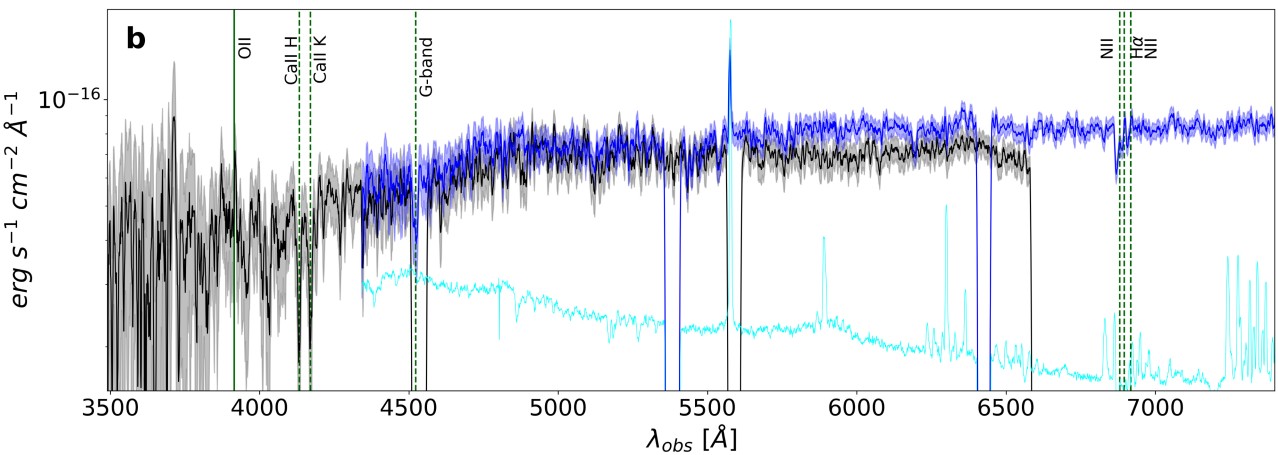

**Extended Data Fig. 1 | eRO-QPE1 position and identification. a**, Legacy DR8 image cut-out around the optical counterpart of eRO-QPE1. Red and green circles represent the astrometry-corrected eROSITA and XMM-Newton EPIC-pn positions, respectively, with 1σ positional uncertainties. The EPIC-pn position was corrected excluding the target (blue cross) to ensure an unbiased estimate of the possible positional offset. Image reproduced from Legacy Surveys / D. Lang (Perimeter Institute) under a CC-BY-4.0 licence. **b**, SALT spectra of eRO-QPE1 shown in black and blue with related 1σ errors as shaded regions. The cyan spectrum represents a re-normalized sky spectrum to guide the eye for the residual sky feature around 5,577 Å.

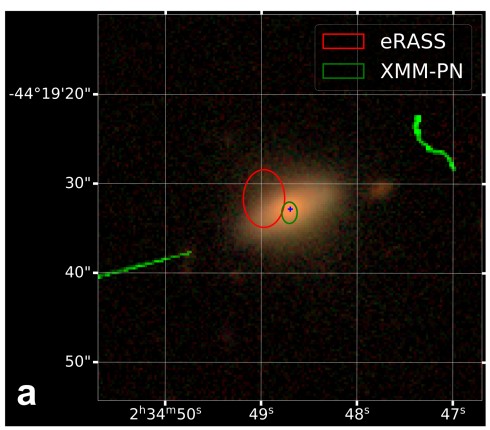

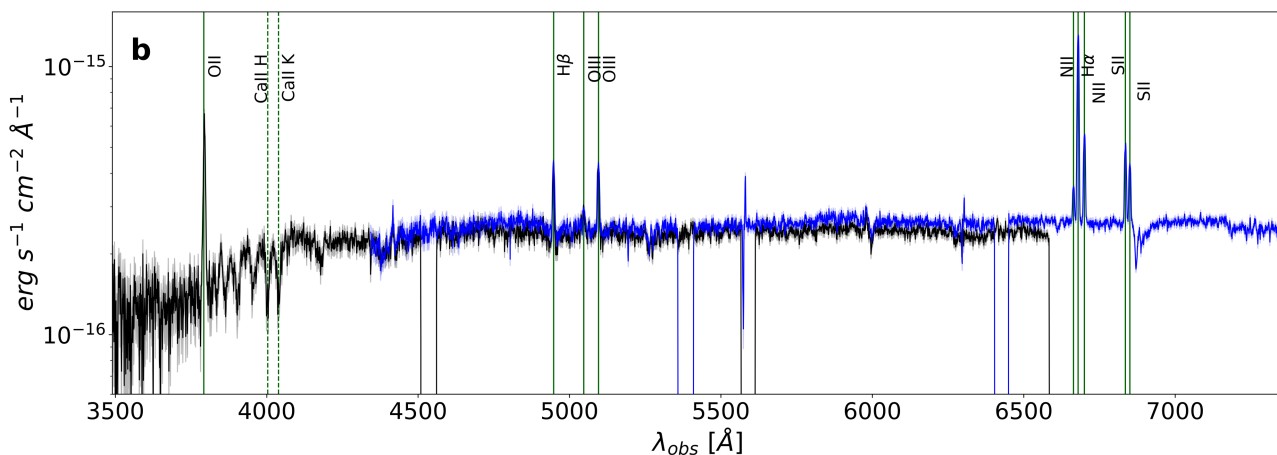

**Extended Data Fig. 2 | eRO-QPE2 position and identification.** As in Extended Data Fig. 1, for eRO-QPE2. Green pixels in **a** are artefacts or missing data in the optical image. Image in **a** reproduced from Legacy Surveys / D. Lang (Perimeter Institute) under a CC-BY-4.0 licence.

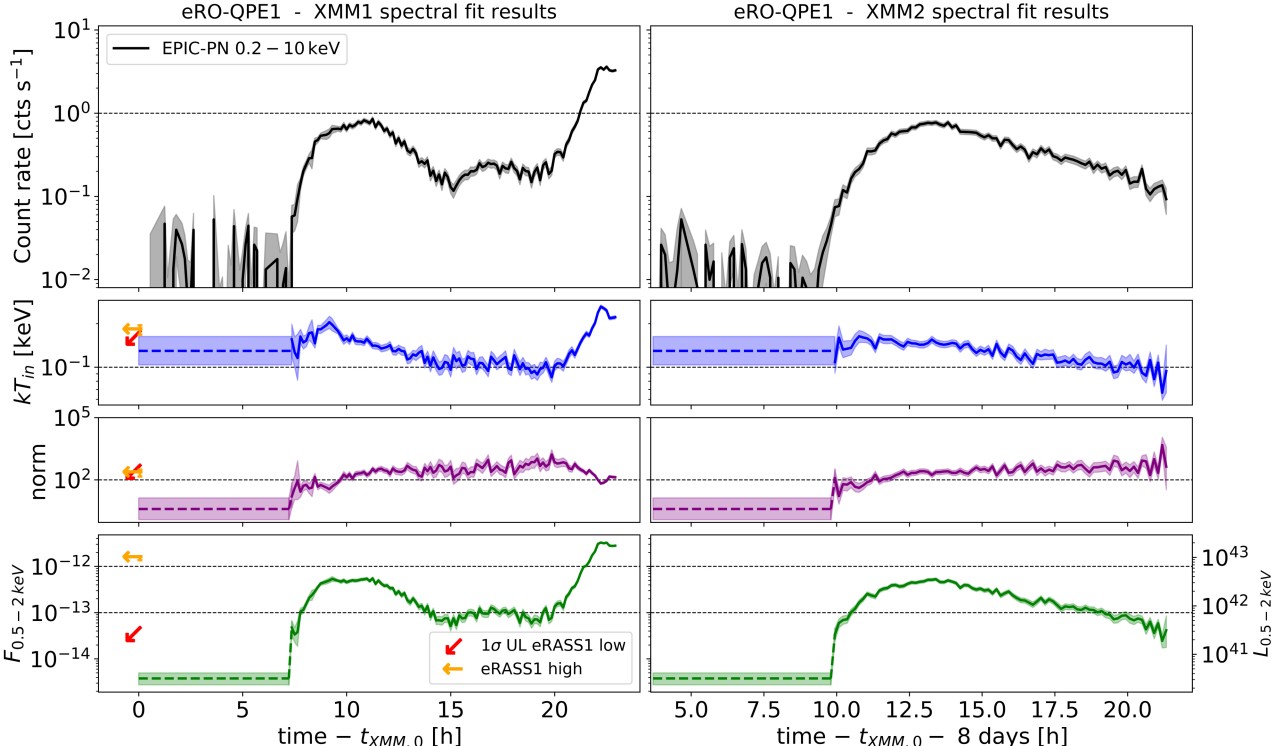

**Extended Data Fig. 3 | eRO-QPE1 spectral fit results.** XMM-Newton EPIC-pn light-curve (top panel) and time-resolved spectroscopy fit results for spectra extracted in the 500-s time bins (bottom panels) of the two XMM-Newton observations of eRO-QPE1 (left, XMM1; right, XMM2) using an accretion disk model (diskbb): in particular, the evolution of the peak accretion disk temperature ($T_{in}$, $k = k_B$) and the normalization (norm), which is proportional to the inner radius once distance and inclination are known. The time evolution of the 0.5–2.0-keV flux ($F_{0.5-2.0keV}$) and luminosity ($L_{0.5-2.0keV}$) is also shown in the bottom panel. The quiescence level is fitted by combining the first part of both XMM-Newton observations. It is shown with a dashed line because, due to low counts, the fit is more uncertain (see Extended Data Fig. 5a). Median fit values and fluxes of the high and low eROSITA states are reported with orange and red arrows pointing left (upper limits (UL) are denoted with diagonal arrows). $1\sigma$ uncertainties on the fit results are shown with shaded regions around the median.

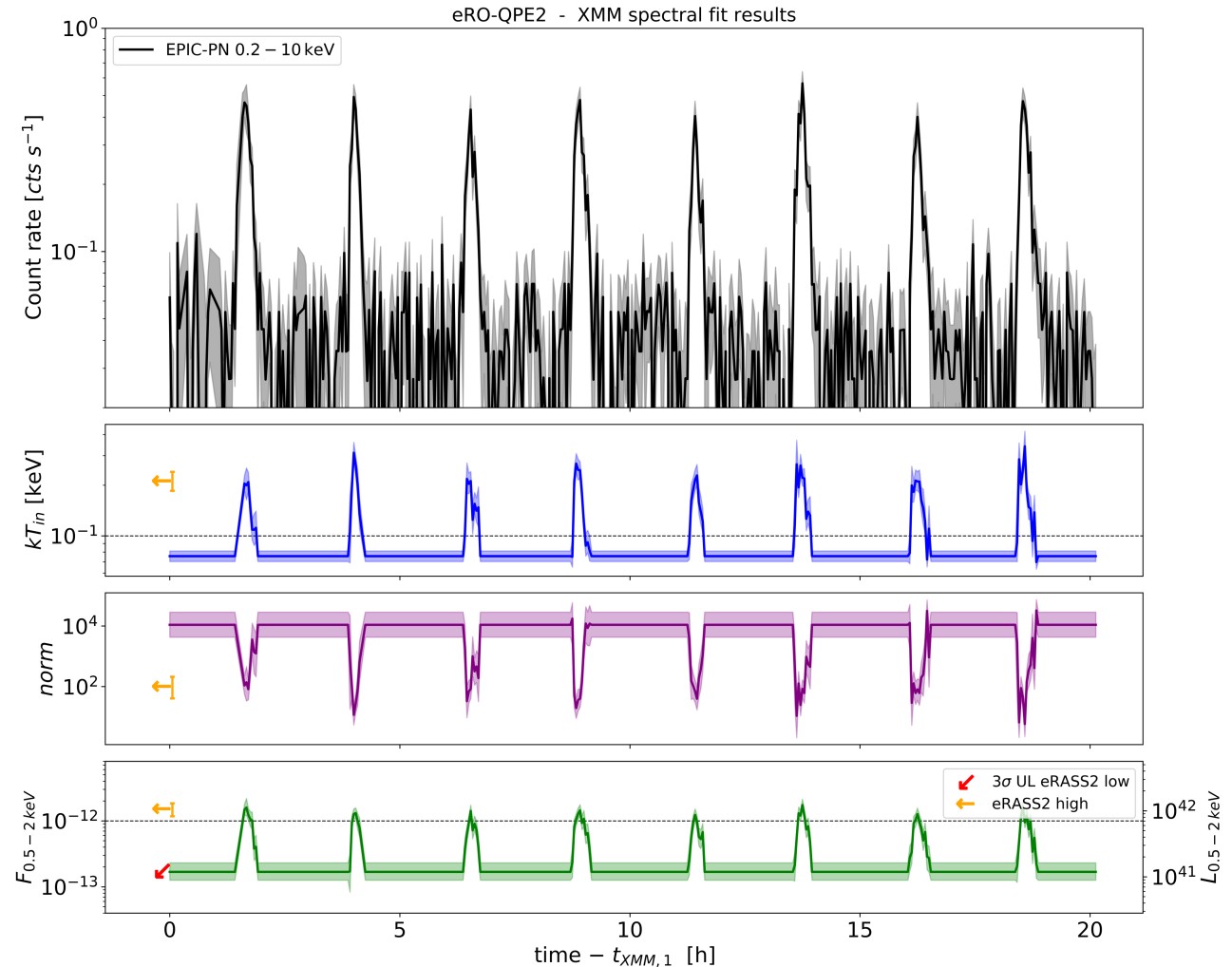

**Extended Data Fig. 4 | eRO-QPE2 spectral fit results.** As in Extended Data Fig. 3, for eRO-QPE2. Here the eROSITA upper limit (UL) of the low state is reported at 3σ.

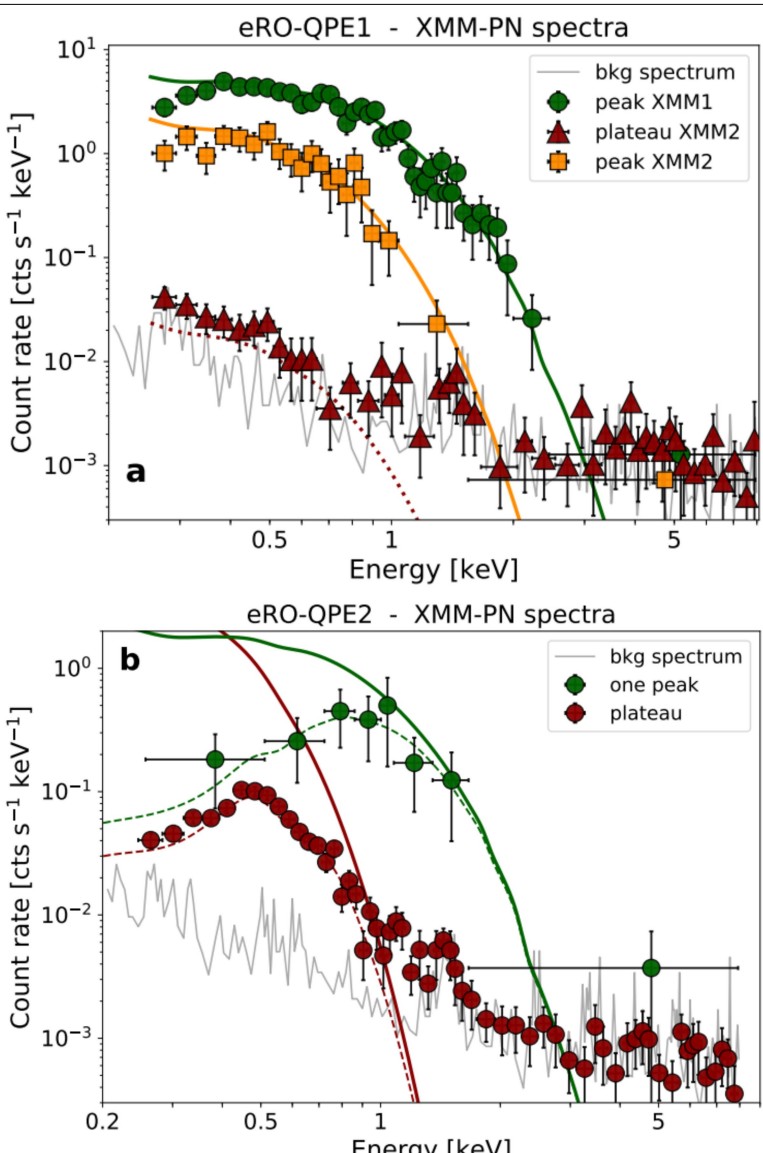

**Extended Data Fig. 5 | eRO-QPE1 and eRO-QPE2 spectra. a**, XMM-Newton EPIC-pn source plus background (bkg) spectra for eRO-QPE1. Red, orange and green data correspond to quiescence and to the peak of the second and first XMM-Newton observations, respectively, with error bars showing 1σ uncertainties. The related solid lines show the unabsorbed source model obtained with diskbb, just for visualization. The grey line represents the background spectrum alone. The plateau is shown with a dotted line because, due to low counts, the fit is more uncertain. **b**, As in **a**, for eRO-QPE2. Here green data represent one of the peaks and the additional dashed lines indicate the absorbed source model.

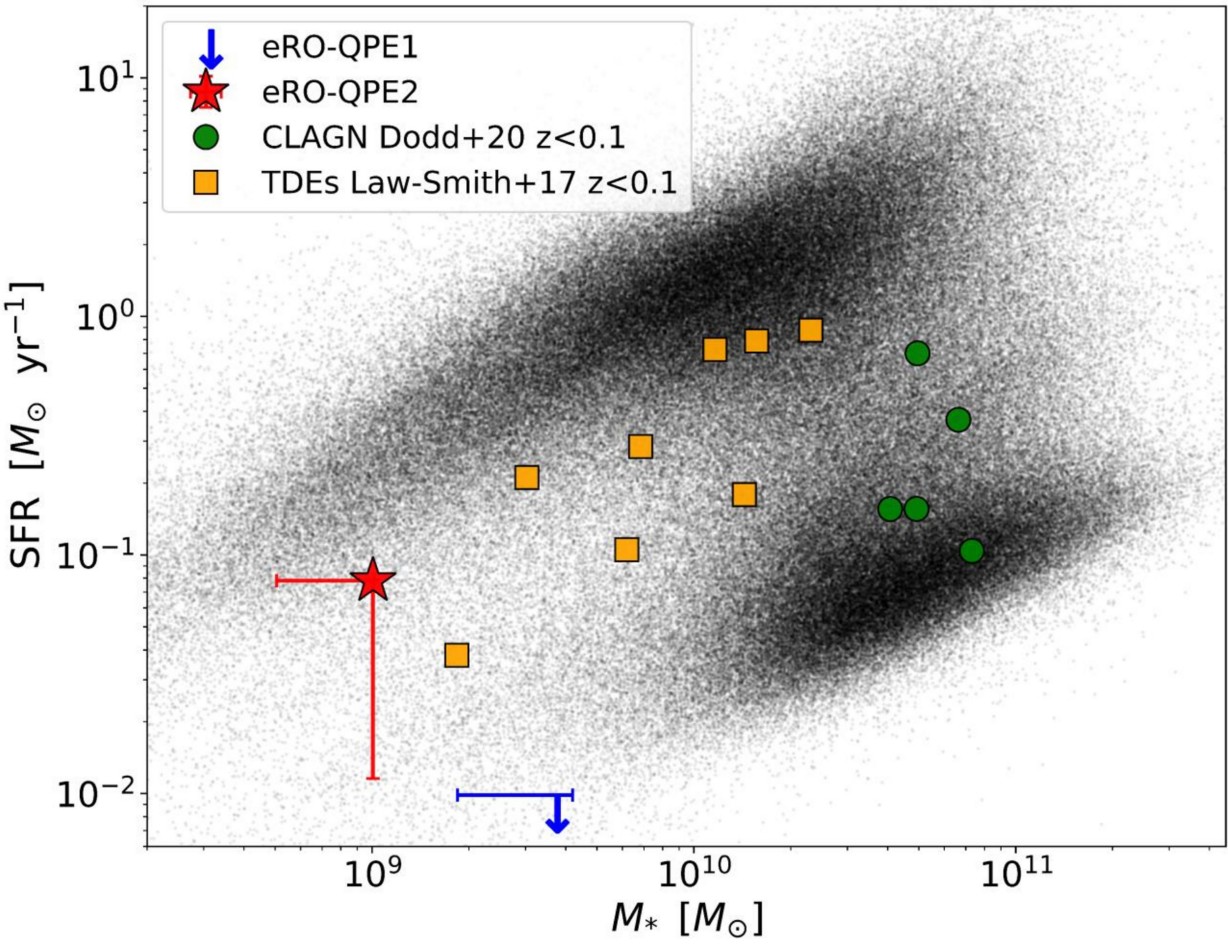

**Extended Data Fig. 6 | The properties of the host galaxies of the QPEs.**
Stellar mass $M_*$ and star-formation rate (SFR) for eRO-QPE1 (blue) and
eRO-QPE2 (red), with related $1\sigma$ uncertainties; for eRO-QPE1, SFR is largely
unconstrained (see Methods section 'The host galaxies of the QPEs'). For a
comparison, normal galaxies[66], TDEs[57] and CLAGN[65], all below $z < 0.1$, are also
shown.

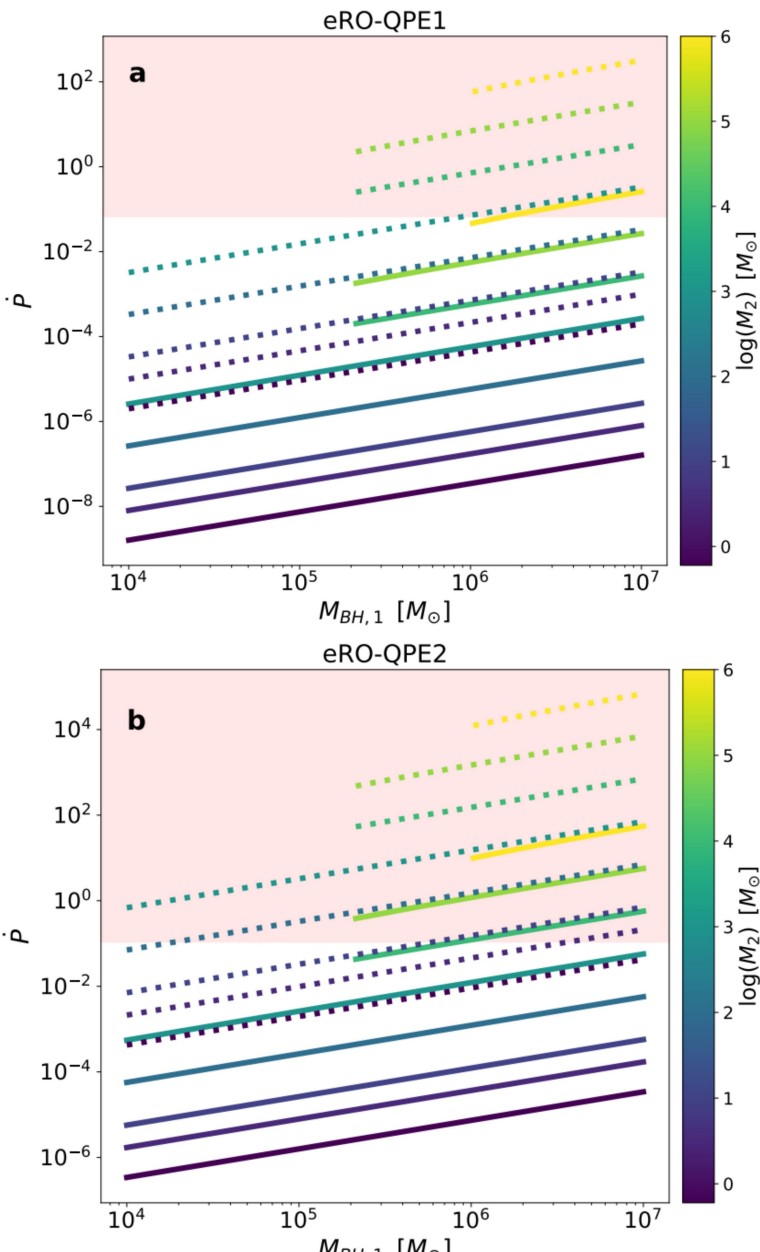

**Extended Data Fig. 7 | Constraints on a secondary orbiting body. a**, Allowed parameter space in terms of the derivative of the period $\dot{P}$ and secondary mass $M_2$ for a range of primary mass $M_{\mathrm{BH,1}} \approx 10^4 M_\odot$–$10^7 M_\odot$ and zero (solid lines) or high orbital eccentricity ($e_0 \approx 0.9$, dotted lines), in which we can reproduce the rest-frame period of eRO-QPE1. We have additionally imposed $M_2 \leq M_{\mathrm{BH,1}}$. We have drawn an approximate threshold at the minimum value of $\dot{P}$ which we would have measured within the available observations, corresponding to a period decrease of one QPE cycle over the 15 observed by NICER (Fig. 1d). The excluded region is shaded in red. **b**, As in **a**, for eRO-QPE2, and adopting as tentative minimum $\dot{P}$ a period decrease of one cycle over the nine observed with XMM-Newton (Fig. 2c).

**Extended Data Table 1 | Summary of the observations performed**

| Source | Instrument | Obs. ID | Start date |
|---|---|---|---|
| **eRO-QPE1** | eROSITA | - | 16 January 2020 |
| | XMM-Newton | 0861910201 | 27 July 2020 |
| | XMM-Newton | 0861910301 | 4 August 2020 |
| | NICER | 3201730103 | 19 August 2020 |
| | SALT | - | 24 September 2020 |
| **eRO-QPE2** | eROSITA | - | 23 June 2020 |
| | XMM-Newton | 0872390101 | 6 August 2020 |
| | SALT | - | 8 September 2020 |

**Extended Data Table 2 | Summary of spectral fit results for eRO-QPE1**

| Observation | $k_B T$ [eV] | $F_{0.5\text{-}2.0\text{ keV}}$ [cgs] | $F_{disk}$ [cgs] | $L_{0.5\text{-}2.0\text{ keV}}$ [cgs] |
|---|---|---|---|---|
| eROSITA low | $\downarrow 160$ | $\downarrow 3.4 \times 10^{-14}$ | $\downarrow 2.4 \times 10^{-13}$ | $\downarrow 2.1 \times 10^{41}$ |
| eROSITA high | $180^{195}_{168}$ | $1.5^{1.7}_{1.4} \times 10^{-12}$ | $4.2^{4.6}_{3.7} \times 10^{-12}$ | $0.9^{1.0}_{0.8} \times 10^{43}$ |
| XMM quiescence | $130^{163}_{103}$ | $3.8^{5.0}_{2.7} \times 10^{-15}$ | $1.9^{2.5}_{1.5} \times 10^{-14}$ | $2.3^{3.1}_{1.7} \times 10^{40}$ |
| XMM1 peak | $262^{269}_{256}$ | $3.3^{3.4}_{3.2} \times 10^{-12}$ | $6.4^{6.6}_{6.2} \times 10^{-12}$ | $2.0^{2.1}_{1.9} \times 10^{43}$ |
| XMM2 peak | $148^{156}_{141}$ | $5.3^{5.6}_{4.9} \times 10^{-13}$ | $2.0^{2.1}_{1.8} \times 10^{-12}$ | $3.2^{3.5}_{3.0} \times 10^{42}$ |

The median value and related 16th and 84th percentile values are reported for every quantity; for unconstrained values 1σ upper limits are quoted using the 84th percentile value of the parameter posterior distribution and are denoted with ↓. Reported results are obtained with the model tbabs convoluted with diskbb, with Galactic $N_H$ frozen at $2.23 \times 10^{20}$ cm$^{-2}$, as reported by the HI4PI Collaboration[50]. Fluxes and luminosities are unabsorbed and rest-frame. The two eROSITA states are shown in Fig. 1a, and the three XMM-Newton observations in the table correspond to the three spectra in Extended Data Fig. 5a. The disk flux ($F_{disk}$) is computed within 0.001 and 100 keV.

**Extended Data Table 3 | Summary of spectral fit results for eRO-QPE2**

| Observation | $N_H(z)$ [cm$^{-2}$] | $k_B T$ [eV] | $F_{0.5-2.0\ keV}$ [cgs] | $F_{disk}$ [cgs] | $L_{0.5-2.0\ keV}$ [cgs] |
|---|---|---|---|---|---|
| eROSITA low | $0.32^{0.38}_{0.28} \times 10^{22}$ | - | $\downarrow 5.7 \times 10^{-14}$ | $\downarrow 3.4 \times 10^{-13}$ | $\downarrow 4.0 \times 10^{40}$ |
| eROSITA high | $0.32^{0.37}_{0.28} \times 10^{22}$ | $209^{241}_{185}$ | $1.5^{1.8}_{1.2} \times 10^{-12}$ | $3.3^{4.5}_{2.4} \times 10^{-12}$ | $1.0^{1.3}_{0.8} \times 10^{42}$ |
| XMM quiescence | $0.35^{0.40}_{0.30} \times 10^{22}$ | $76^{81}_{70}$ | $1.7^{2.3}_{1.3} \times 10^{-13}$ | $8.0^{14.0}_{4.5} \times 10^{-13}$ | $1.2^{1.6}_{0.9} \times 10^{41}$ |
| XMM peak | $0.33^{0.39}_{0.30} \times 10^{22}$ | $222^{249}_{199}$ | $1.7^{2.1}_{1.5} \times 10^{-12}$ | $9.1^{14.5}_{4.3} \times 10^{-12}$ | $1.2^{1.5}_{1.0} \times 10^{42}$ |

As in Extended Data Table 1, for eRO-QPE2. Reported results are obtained with the model tbabs × ztbabs × diskbb (where × represents a convolution), with Galactic $N_H$ frozen at $1.66 \times 10^{20}$ cm$^{-2}$, as reported by the HI4PI Collaboration[50]; absorption in excess was estimated from 'XMM quiescence' and was allowed to vary within its 10th and 90th percentiles for all the other observations. The two eROSITA states are shown in Fig. 2a and model parameters in the low state are unconstrained; the two XMM-Newton observations in the table correspond to the spectra in Extended Data Fig. 5b.