## [Peer Review File · Nature]

Supplementary information

X-ray quasi-periodic eruptions from two previously quiescent galaxies

In the format provided by the authors and unedited

Peer Review File

Manuscript Title: X-ray Quasi-Periodic Eruptions from two previously quiescent galaxies

Reviewer Comments & Author Rebuttals**Reviewer Reports on the Initial Version:**

Referees' comments:

Referee #1 (Remarks to the Author):

This paper presents fascinating new evidence related to a recently-discovered phenomenon, quasi-periodic eruptions (QPEs) from galactic nuclei. The two newly-discovered QPEs double the previous sample and extend it in several interesting ways. I am happy to recommend publication in Nature after several observational and theoretical issues have been addressed.

Observational issues:

As demonstrated in Figures 1 and 2, the near-periodicities of the eruptions in QPE1 and QPE2 are well-established over a short time using NICER (for QPE1) and XMM-Newton (for QPE2) data. Have there been additional NICER and XMM observations in the succeeding 3-4 months? If so, the additional data need to be displayed. If not, why not? If the targets have been available for observation (e.g., if they have not been in a Sun avoidance zone) but have not been observed since, then this is a major missed opportunity. The interesting suggestion that these eruptions result from partial tidal disruption of a white dwarf has a number of associated predictions that could be far better tested with a baseline of 3-4 months than with a baseline of 10 days. These include the overall periodicity of the source and possible secondary effects such as the decay of the orbit due to gravitational radiation and the possibility of precession of the orbit. Continuous observation is not necessary; a procedure similar to phase connection in pulsar astronomy could be used.

Along those lines, is it possible to bring together all the QPE1, and separately all the QPE2, observations to attempt phase connection?

Neither source had been seen before. More detail about previous non-detections would be useful. For example, given the duty cycle and distribution of fluxes during the current active period, previous observations of this field could be used to establish, quantitatively, a probability that the sources have turned on in the last few years rather than simply having been missed. This matters because, for example, the postulated highly eccentric orbits have a very short lifetime (both gravitational radiation and tidal effects will reduce the eccentricity).

On line 563 we read that there are longer separations after more intense bursts, with a reference to the timing analysis section. This is potentially an important clue to the nature of the bursts, but it is not mentioned in the timing analysis section. Details should be provided.

Theoretical issues:

One of the main conclusions of this paper is that the limit-cycle hypothesis is disfavored whereas a binary origin is promising. That may be, but the comparison between the two models is not on level ground. The limit-cycle model is specific and well developed, and it has clear expectations. The binary model has not been developed nearly as much, which means that it is not easy to

compare the data with specific predictions. For example, would we expect that matter donated in previous cycles would build up in the system and therefore that we would expect low-level continuous emission in addition to the bursts? Are the rates of such events reasonable? Should we expect no detected simultaneous optical and UV variability, or is that a problem for the model? Is the theoretical spectrum consistent with the observations? With all of this and more, and given that there might be additional twists to the limit-cycle model, the conclusions need to be softened (although, personally, I'd say that the binary model is probably right). For example, the abstract could read "Indeed, the periods... are inconsistent with current models that invoke radiation-pressure accretion disk instabilities." This makes it clear that modifications to those models are in play, at least in principle. Similar caution should be injected into the detailed discussion of limit-cycle models later in the text.

Another major conclusion of this paper is that QPEs are viable candidates for EM counterparts of extreme mass ratio inspirals. I would like the authors to make one simple change in their wording in the abstract (and associated other places in the text): change "This scenario would make QPEs a viable candidate" to "This scenario could make QPEs a viable candidate". The reason is that this would be a wonderful possibility, but there are reasons to think that they will be extremely uncommon. For example, if the secondary is a 0.2 Msun white dwarf (as has been suggested for GSN 069) then the gravitational wave amplitude around a given primary is 2% of what it would be for a 10 Msun black hole secondary. Thus for a fixed signal to noise over a given observational time, the 0.2 Msun white dwarf system would have to be 50x closer, i.e., within a volume just 10^{-5} of the volume in which the 10 Msun black hole system would be visible. Most EMRIs will not be of this type. There are also concerns about the viability for other types of stars. But putting in "could" will make it clear that this is not definite.

In lines 129-131 we read about constraints based on the emission of gravitational waves. One point to keep in mind is that if the orbit consists of a secondary which is partially stripped of mass at each pericenter passage, then the mass loss will produce a kick on the orbit. Thus the orbital evolution will not be exclusively due to gravitational radiation. Also, note that if the secondary is a white dwarf (as has been suggested for GSN 069) then it cannot be tidally stripped by a 10^7 Msun black hole because the white dwarf is dense enough to remain intact even at the event horizon. The upper limit is a bit over 10^6 Msun for such a white dwarf, depending on the spin of the supermassive black hole.

A key observation not discussed in the context of the orbiting body idea is the lack of simultaneous UV and/or optical variability. This has to be addressed. Surely one would expect *some* variability. Is there a plausible reason why we couldn't see it in this case?

Referee #2 (Remarks to the Author):

The authors reported two quasi-periodic eruptions (QPEs) discovered by the eROSITA survey and their follow-up observations and analyses. The discoveries, eRO-QPE1 and eRO-QPE2, were triggered by a systematic search during the first two all-sky scans, and their follow-up observations with NICER and XMM detected ~ 18 and 9 further bursts of the sources, respectively, revealing their remarkable quasi-periodic nature. There are two main possible origins of QPEs: from the inner accretion flow and driven by instability, or due to a secondary orbiting body. The former is disfavored by the authors. They instead suggest that QPEs could be the electromagnetic counterparts to extreme mass ratio inspirals (EMRIs), one type of highly-anticipated gravitational wave sources for LISA.

The paper is interesting in several ways: 1) It is an interesting discovery enabled by a new survey. Previously, QPEs were found serendipitously or in an archival search; with the eROSITA survey, the whole X-ray sky (half, in this case) has been opened up for discovery through a systematic search. This is also a very timely study: QPEs are a new phenomenon discovered in the last \sim year.

2) If QPEs are indeed the EM counterparts of EMRIs, then there is an exciting prospect for multi-messenger astrophysics with LISA, so this study will be of interest to observational astronomers and GW astrophysicists in the broader community.

My main comment is that eRO-QPE1 strikes me as different from eRO-QPE2 and the previous QPEs (GSN 069 and RX J1301.9+2747), even though it shares some common features with them, such as a short quasi-period and very little UV/optical variability. 1) Its temperature in the cold/faint phase ("XMM quiescence") is quite warm and comparable to the warm/bright phase ("XMM2 peak"), and it's unclear whether there is an emergence of a soft excess-like component in the bright phase, which is a key characteristic in GSN 069 and RX J1301. The authors fitted both faint and bright phases to zbody or diskbb. Have they tried applying a two-component spectral fit to the bright phase? 2) The light curve profile is asymmetric and broad (reminiscent of a TDE), and overall, does not look like a scaled version of the other QPEs. Both objects are interesting in the context of EMRI EM counterpart, but I'm a little skeptical whether eRO-QPE1 is a bona fide, GSN 069-like QPE.

Some minor comments:

All multi-panel figures: the labels a, b, ... are not marked on the figures themselves.

Lines 49-50: "...if the X-ray spectra are with a standard accretion disk model..."
Consistent with?

Lines 102-103: "...our blind search is inherently designed to sample the QPE population without bias..."

Please clarify that their search is less biased with respect to the host galaxies, not the QPEs (the ~4 hr sampling is a bias, for example).

Lines 138-141 (also: lines 552 -570, Extended Data Fig. 7): By definition, they could additionally impose $M_1 > M_2$.

Lines 455-457: Does a scaling relation apply here? If QPEs are due to an orbiting body, then a clear correlation between quasi-period and black hole mass should not be expected (unlike the standard QPOs).

The section starting from line 534: there are a few references to "dual" AGN, but some of them are actually binaries. In other places, they refer to the same type of system (correctly) as a "binary" SMBH, so the authors know they are different objects but are being inconsistent with the nomenclature.

Same section: the authors may also want to mention that if the periodic variability is produced by the beaming effects of the binary minidisks, the modulation should be smooth and quasi-sinusoidal, in contrast to the observed QPEs. They could also discuss, at the qualitative level, whether other mechanisms such as modulated accretion variability and self-lensing (which could produce sharp peaks) are plausible. (Though the best arguments against the binary scenario are what they said in the next paragraph, i.e. negative \dot{P} and implied rates.)

I think the authors should also discuss whether other mechanisms, such as precession, could produce the observed QPEs.

I also want to encourage the authors to do their best to better reference relevant previous literature. For example, references on supermassive black hole binaries in this section could be more complete and/or appropriate.

Author Rebuttals to Initial Comments:

NOTE: Author rebuttals in RED

Referee #1 (Remarks to the Author):

This paper presents fascinating new evidence related to a recently-discovered phenomenon, quasi-periodic eruptions (QPEs) from galactic nuclei. The two newly-discovered QPEs double the previous sample and extend it in several interesting ways. I am happy to recommend publication in Nature after several observational and theoretical issues have been addressed.

Observational issues:

As demonstrated in Figures 1 and 2, the near-periodicities of the eruptions in QPE1 and QPE2 are well-established over a short time using NICER (for QPE1) and XMM-Newton (for QPE2) data. Have there been additional NICER and XMM observations in the succeeding 3-4 months? If so, the additional data need to be displayed. If not, why not? If the targets have been available for observation (e.g., if they have not been in a Sun avoidance zone) but have not been observed since, then this is a major missed opportunity. The interesting suggestion that these eruptions result from partial tidal disruption of a white dwarf has a number of associated predictions that could be far better tested with a baseline of 3-4 months than with a baseline of 10 days. These include the overall periodicity of the source and possible secondary effects such as the decay of the orbit due to gravitational radiation and the possibility of precession of the orbit.

Regarding the need for a longer baseline to test the proposed scenario, we fully agree with Referee #1. Regarding additional data, this first round of “discovery” observations was of course limited, but we did already ask for more data (QPE1) or we will in the next months (QPE2). For instance, we have asked NICER data for QPE1 to be taken both in Summer 2021 and early 2022, although not in DDT time but regular observing time, and the panel has not evaluate proposals yet.

Continuous observation is not necessary; a procedure similar to phase connection in pulsar astronomy could be used. Along those lines, is it possible to bring together all the QPE1, and separately all the QPE2, observations to attempt phase connection?

Our light curve analysis consisted in simultaneously fitting all the bursts independently with in addition some constant emission for the plateau. This method shares all the advantages of folding a light curve in terms of finding the different peaks, how different the bursts are, estimating rise and decay durations, and so on. We share here for Referee#1 a folded profile of eRO-QPE1 with NICER data as example.

We think that we do not gain more information with respect to the light curve fitting method (which accounts for the shape of each bursts independently) and we chose to report only the details about the timing analysis that were necessary for what is discussed in the paper (e.g. ‘On accretion flow instabilities’ and ‘On the presence of an orbiting body’). For instance, details of a single burst for eRO-QPE1, representative of all, are already in the paper and could be seen in the second XMM ToO in log scale.

Neither source had been seen before. More detail about previous non-detections would be useful. For example, given the duty cycle and distribution of fluxes during the current active period, previous observations of this field could be used to establish, quantitatively, a probability that the sources have turned on in the last few years rather than simply having been missed. This matters because, for example, the postulated highly eccentric orbits have a very short lifetime (both gravitational radiation and tidal effects will reduce the eccentricity).

In the original manuscript we had included a couple of sentences about ROSAT and XMM-Slew upper limits in ‘X-ray spectral analysis’, but it was probably the wrong place. We have now combined this and expanded at the end of section ‘Blind search for QPEs with eROSITA’ in a paragraph called ‘Previous X-ray activity’. Unfortunately the duration of the ROSAT and XMM-Slew observation is so short (specially compared with the QPEs’ evolving timescales and related dispersions) that one cannot obtain any meaningful constraint on the start of QPEs. This is however consistent also with QPEs starting only recently, which would not invalidate the model.

As for data from other wavelengths, there seems to be hardly any in archival catalogs (we have checked Vizier and some additional known catalogs). These sources were unknown at most wavelengths. The exception was some optical-IR, occasionally UV, photometry which we used for an early photo-z estimate, which was then superseded by our new optical spectra. We realize we had not written this anywhere, therefore we added a sentence at the beginning of ‘The host galaxies of

QPEs'.

On line 563 we read that there are longer separations after more intense bursts, with a reference to the timing analysis section. This is potentially an important clue to the nature of the bursts, but it is not mentioned in the timing analysis section. Details should be provided.

We sincerely apologize for this inconsistency. At a late stage of the draft we had to trim the text down and we shortened a bit the 'Timing analysis' section. But most importantly, that sentence was meant to be removed also from line 563 because we cannot securely state whether that is the case or not. With an early quick analysis it seemed that there was a relation between fluence and period, but after doing more appropriate checks it was deemed not significant. That is why we decide not to claim anything explicitly about this. More bursts from future observation will help in obtaining a more stringent constraint and we will keep performing this analysis in the future, but we cannot claim anything now. That sentence was forgotten in the article, we now remove it.

Theoretical issues:

One of the main conclusions of this paper is that the limit-cycle hypothesis is disfavored whereas a binary origin is promising. That may be, but the comparison between the two models is not on level ground. The limit-cycle model is specific and well developed, and it has clear expectations. The binary model has not been developed nearly as much, which means that it is not easy to compare the data with specific predictions. For example, would we expect that matter donated in previous cycles would build up in the system and therefore that we would expect low-level continuous emission in addition to the bursts? Are the rates of such events reasonable? Should we expect no detected simultaneous optical and UV variability, or is that a problem for the model? Is the theoretical spectrum consistent with the observations? With all of this and more, and given that there might be additional twists to the limit-cycle model, the conclusions need to be softened (although, personally, I'd say that the binary model is probably right). For example, the abstract could read "Indeed, the periods... are inconsistent with current models that invoke radiation-pressure accretion disk instabilities." This makes it clear that modifications to those models are in play, at least in principle. Similar caution should be injected into the detailed discussion of limit-cycle models later in the text.

Another major conclusion of this paper is that QPEs are viable candidates for EM counterparts of extreme mass ratio inspirals. I would like the authors to make one simple change in their wording in the abstract (and associated other places in the text): change "This scenario would make QPEs a viable candidate" to "This scenario could make QPEs a viable candidate". The reason is that this would be a wonderful possibility, but there are reasons to think that they will be extremely uncommon. For example, if the secondary is a 0.2 Msun white dwarf (as has been suggested for GSN 069) then the gravitational wave amplitude around a given primary is 2% of what it would be for a 10 Msun black hole secondary. Thus for a fixed signal to noise over a given observational time, the 0.2 Msun white dwarf system would have to be 50x closer, i.e., within a volume just 10^{-5} of the volume in which the 10 Msun black hole system would be visible. Most EMRIs will not be of this type. There are

also concerns about the viability for other types of stars. But putting in "could" will make it clear that this is not definite.

We answer to both points above since they both share the purpose of re-shaping the strength of our claims. We fully agree with Referee #1 on all aspects of the comments, it is indeed true that the two models are not equally developed, and that our data are inconsistent with *current* instability models. We have now changed the related sentences in the summary (lines 29-30 and 34); then we also changed part of the text of the main article, e.g. line 129 to stress that our scenario is meant to be simplistic; we also change some words accordingly elsewhere and in the Methods ('On accretion flow instabilities' and 'On the presence of an orbiting body').

In lines 129-131 we read about constraints based on the emission of gravitational waves. One point to keep in mind is that if the orbit consists of a secondary which is partially stripped of mass at each pericenter passage, then the mass loss will produce a kick on the orbit. Thus the orbital evolution will not be exclusively due to gravitational radiation. Also, note that if the secondary is a white dwarf (as has been suggested for GSN 069) then it cannot be tidally stripped by a 10^7 Msun black hole because the white dwarf is dense enough to remain intact even at the event horizon. The upper limit is a bit over 10^6 Msun for such a white dwarf, depending on the spin of the supermassive black hole.

We thank Referee #1 for the clarifications. We realize we have been too simplistic and incomplete with our wording. In the current text we attempt to correct this while keeping the text as smooth as possible: we changed the phrasing of the paragraph starting at line 144 and some words of the Methods ('On the presence of an orbiting body') accordingly.

A key observation not discussed in the context of the orbiting body idea is the lack of simultaneous UV and/or optical variability. This has to be addressed. Surely one would expect *some* variability. Is there a plausible reason why we couldn't see it in this case?

We thank Referee #1 for the input, we have now added our answer to this question at the end of the section 'On the presence of an orbiting body' in the Methods, before the start of the paragraph '*Predicted numbers*'.

Referee #2 (Remarks to the Author):

The authors reported two quasi-periodic eruptions (QPEs) discovered by the eROSITA survey and their follow-up observations and analyses. The discoveries, eRO-QPE1 and eRO-QPE2, were triggered by a systematic search during the first two all-sky scans, and their follow-up observations with NICER and XMM detected ~ 18 and 9 further bursts of the sources, respectively, revealing their remarkable quasi-periodic nature. There are two main possible origins of QPEs: from the inner accretion flow

and driven by instability, or due to a secondary orbiting body. The former is disfavored by the authors. They instead suggest that QPEs could be the electromagnetic counterparts to extreme mass ratio inspirals (EMRIs), one type of highly-anticipated gravitational wave sources for LISA.

The paper is interesting in several ways: 1) It is an interesting discovery enabled by a new survey. Previously, QPEs were found serendipitously or in an archival search; with the eROSITA survey, the whole X-ray sky (half, in this case) has been opened up for discovery through a systematic search. This is also a very timely study: QPEs are a new phenomenon discovered in the last ~year. 2) If QPEs are indeed the EM counterparts of EMRIs, then there is an exciting prospect for multi-messenger astrophysics with LISA, so this study will be of interest to observational astronomers and GW astrophysicists in the broader community.

My main comment is that eRO-QPE1 strikes me as different from eRO-QPE2 and the previous QPEs (GSN 069 and RX J1301.9+2747), even though it shares some common features with them, such as a short quasi-period and very little UV/optical variability.

1) Its temperature in the cold/faint phase (“XMM quiescence”) is quite warm and comparable to the warm/bright phase (“XMM2 peak”), and it’s unclear whether there is an emergence of a soft excess-like component in the bright phase, which is a key characteristic in GSN 069 and RX J1301.

We agree about this statement, but we want to draw the Referee#2’s attention to the plateau spectrum in Fig. 5a. We want to stress that an artificial hardening of the source’s diskbb might have occurred due to background subtraction in a background-dominated part of the spectrum. It is quite typical for low counts soft X-ray spectra to show apparently decent fits which yield “harder” results, with very little possibility to quantify this conclusively. The plateau of eRO-QPE2 is instead much brighter in flux (merely due to the redshift being 0.0175 instead of 0.0505), so this might be contributing to the difference. Moreover, this is the reason why for eRO-QPE1 the plateau line is dashed in this figure, and dotted in Ext. Data Fig. 5a. We wanted to draw caution, but we must have forgotten to add a note in the figures caption. Now added.

The authors fitted both faint and bright phases to zbody or diskbb. Have they tried applying a two-component spectral fit to the bright phase?

We did not so far, mostly to keep the article as short as possible, given the already high number of sections and figures in the Methods. Moreover, in our opinion a multi-component fit in GSN 069 and RX J1301 in the respective papers was done to test whether QPEs were related to the formation of a soft-excess in an unstable inner disk region. In our paper however, we do not follow the same line of reasoning, therefore we thought that more complex spectral modeling was redundant for our discussion. Of course, a comparison of all 4 QPEs, both with timing and spectral data, is of great interest for a future work.

For completeness, we share here results of a 2-components fit for eRO-QPE2, in terms of temperature of the second component. We show, for each spectrum of any non-plateau region, the temperature of the additional zbody on top of a diskbb-plateau, compared with the results of the single diskbb alone shown in our paper ($kT_{in,1disk}$). The small offset ($kT_{bb,2comp}$ is slightly lower) is due to the fact that of course the 2-component fit has an underlying plateau. But other than that results are consistent within uncertainties, as they should be by definition of those models (diskbb itself is already a superposition of black bodies).

2) The light curve profile is asymmetric and broad (reminiscent of a TDE), and overall, does not look like a scaled version of the other QPEs. Both objects are interesting in the context of EMRI EM counterpart, but I'm a little skeptical whether eRO-QPE1 is a bona fide, GSN 069-like QPE.

We agree that the phenomenology of the bursts in eRO-QPE1 seems different, but we are convinced it is merely because of the much larger evolving timescales. From a practical point of view, with 4 objects we do not know yet what is typical or not, nor we can expect them to be all the same in every detail (the inclination for instance must have some impact). But we agree that the asymmetry, if only found in one source even in the future with much larger samples, might be puzzling. However, we want to draw the attention of Referee#2 to the QPEs of the source published by Giustini et al., (2020). We display here the second burst of the 2019 data, just as an example, first with data only (left), then an asymmetric profile (red) and a Symmetric (orange) on the right:

The point we want to make is not that we believe these bursts to be securely asymmetric, in fact both fit are equally good (perhaps the red model on the rise of the burst is better?). But it looks like we cannot exclude that QPEs in the three other sources look symmetric mostly due to their much shorter durations. Effectively, squeezing an asymmetric profile to a tenth of the duration of eRO-QPE1 would make both models equally good (as we tried for eRO-QPE2 and we show here); therefore for the shorter QPEs it makes sense to keep the symmetric one for simplicity.

Our point is that, at this stage of our knowledge we would consider all four of them QPE sources in the same ball-park.

Some minor comments:

We thank Referee#2 for all this minor comments. If we do not comment below every item, it means we applied the suggested change as is.

All multi-panel figures: the labels a, b, ... are not marked on the figures themselves.

Lines 49-50: "...if the X-ray spectra are with a standard accretion disk model..."

Consistent with?

Lines 102-103: "...our blind search is inherently designed to sample the QPE population without bias..."

Please clarify that their search is less biased with respect to the host galaxies, not the QPEs (the ~4 hr sampling is a bias, for example).

Lines 138-141 (also: lines 552 -570, Extended Data Fig. 7): By definition, they could additionally impose $M_1 > M_2$.

Lines 455-457: Does a scaling relation apply here? If QPEs are due to an orbiting body, then a clear correlation between quasi-period and black hole mass should not be expected (unlike the standard QPOs).

We were quite cryptic in that sentence, now we expanded. That scaling relation was intended as the host galaxy- black hole relation to obtain the black hole mass from the host galaxy's stellar mass. It was not intended as the M obtained from X-ray variability.

[The section starting from line 534: there are a few references to "dual" AGN, but some of them are actually binaries. In other places, they refer to the same type of system (correctly) as a "binary" SMBH, so the authors know they are different objects but are being inconsistent with the nomenclature.

Same section: the authors may also want to mention that if the periodic variability is produced by the beaming effects of the binary minidisks, the modulation should be smooth and quasi-sinusoidal, in contrast to the observed QPEs. They could also discuss, at the qualitative level, whether other mechanisms such as modulated accretion variability and self-lensing (which could produce sharp peaks) are plausible. (Though the best arguments against the binary scenario are what they said in the next paragraph, i.e. negative \dot{P} and implied rates.)

I also want to encourage the authors to do their best to better reference relevant previous literature. For example, references on supermassive black hole binaries in this section could be more complete and/or appropriate.

Combining the above comments. We have now expanded a few sentences in the section. We also sincerely apologize for our sloppiness in referencing. We did not exclude any meaningful contribution on purpose. Hopefully now the referencing is improved. In case credit is still not properly given (e.g. if we missed work on a given topic which was key/precursor for that given result/idea), we welcome any advice from Referee#2.

I think the authors should also discuss whether other mechanisms, such as precession, could produce the observed QPEs.

We agree with Referee#2 that precession is worth mentioning. We now added a small paragraph at the end of the Methods section 'On accretion flow instabilities'.

Reviewer Reports on the First Revision:

Referee #1 (Remarks to the Author):

I am satisfied with the changes made by the authors and am now happy to recommend publication.

Referee #2 (Remarks to the Author):

Thanks to the authors for their response. In fact, eRO-QPE1's similarity with the other QPEs is much more clear in the folded light curve the authors shared with Referee #1, so I find it believable that they are indeed the same.

Given the quality of the work, and their interesting results and potential impact, I can certainly see this paper being published in Nature.

Author Rebuttals to First Revision:

N/A